# Inherent instability of simple DNA repeats shapes an evolutionarily stable distribution of repeat lengths

Ryan J. McGinty [1,3], Daniel J. Balick[1,3], Sergei M. Mirkin [2] ✉ & Shamil R. Sunyaev [1] ✉

Using the Telomere-to-Telomere reference, we assemble the distribution of simple tandem repeat lengths present in the human genome. Analyzing over three hundred mammalian genomes, we find remarkable consistency in the shape of the distribution across evolutionary epochs. All observed genomes harbor an excess of long repeats, which are potentially prone to developing into repeat expansion disorders. We measure mutation rates for repeat length instability, quantitatively model the per-generation action of mutations, and observe the corresponding long-term behavior shaping the repeat tract length distribution. We find that short repetitive sequences appear to be a straight-forward consequence of random substitution. Evolving largely independently, longer repeats (above roughly 10 nt) emerge and persist in a rapidly mutating dynamic balance between expansion, contraction, and interruption. These mutational processes, collectively, are sufficient to explain the abundance of long repeats, without invoking natural selection. Our analysis constrains properties of molecular mechanisms responsible for maintaining genome fidelity that underlie repeat instability.

Over 2.5% of human genomic DNA consists of simple DNA repeats[1]. Also known as short tandem repeats (STRs) or microsatellites, simple repeats consist of direct tandem repetitions of short sequence motifs, e.g., mononucleotides, dinucleotides, trinucleotides, and so forth. In a randomized DNA sequence, the probability of encountering a simple repeat is exponentially decreased with increasing tract length. Yet this relationship fails to predict the enormous overrepresentation of long simple repeats in most genomic sequences, including in humans[2–4]. The origin of this abundance remains to be elucidated.

This overrepresentation is even more striking in light of the existence of repeat expansion disorders, a growing list of severe human diseases caused by disruption of gene function due to long STRs[5,6]. Decades of study have demonstrated that repeat tract lengths vary between and within individuals[7], owing to frequent expansion and contraction mutations. The rate of these mutations increases with the length of a repeat, a phenomenon known as repeat length instability[8].

Length instability is commonly ascribed to DNA strand slippage during replication and/or DNA repair, although a variety of other molecular mechanisms can also contribute[8]. Instability rates differ between various repeat motifs, particularly for motifs that form non-B DNA secondary structures[9]. Importantly, when repeat length exceeds a threshold of approximately 75–90 nt, carriers frequently transmit a substantially longer repeat to the next generation. Known as 'genetic anticipation', this effect continues to compound in subsequent generations, which leads to more severe presentation and/or earlier age of onset[6]. Recently developed techniques, such as ExpansionHunter[10] and long-read sequencing, have accelerated the discovery of pathogenic repeats; in particular, the growing number of repeat expansion disorders mapped to introns and other non-coding regions sheds light on repeat disease biology beyond coding regions. Repeat expansions are also observed in various cancers[11–13] and serve as hotspots for genomic rearrangements[14].

[1]Department of Biomedical Informatics, Harvard Medical School, Boston, MA, USA. [2]Department of Biology, Tufts University, Medford, MA, USA. [3]These authors contributed equally: Ryan J. McGinty, Daniel J. Balick. ✉e-mail: Sergei.Mirkin@tufts.edu; ssunyaev@hms.harvard.edu

While numerous studies focus on the instability of disease-length repeats, comparatively less is known about shorter repeats, including the so-called 'long-normal' alleles that sit immediately below the disease-length threshold. Carriers of long-normal repeat alleles are healthy, but risk transmitting a disease-length allele due to the higher rate of repeat expansion; additionally, some long-normal alleles contain protective interruptions that, if lost, result in reversion to disease length[6]. Complementing our understanding of long disease-causing repeats, a recent finding identified an autosomal dominant thyroid disorder linked to a $(TTTG)_4$ repeat, with a recurrent deletion to $(TTTG)_3$ in affected individuals[15]. Additionally, instability of $A_8$ and $C_8$ repeats in the coding sequences of mismatch repair (MMR) genes *MSH3* and *MSH6*, respectively, promotes tumor adaptability via frequent frameshifts and subsequent reversions[16]. The latter examples suggest that relatively short repeats, which comprise a much larger portion of the genome, also have biomedical relevance.

In light of the rapidly growing list of repeat-associated diseases, it is surprising to find repeats harbored in abundance in the genome. Interest in this discrepancy goes back at least three decades[2] and has led to speculation that natural selection preserves longer repeat lengths, despite the risk of disease[17]. The best-supported examples of functionality are specific to telomeric and centromeric repeats[9,17,18], though some recent studies have suggested that simple repeats play a role in gene regulation[17]. However, before assuming the overabundance of repeats is evidence of functionality, a more basic explanation should be considered: the excess of repeats in the genome is solely a consequence of mutational processes. Several studies, largely pre-dating the human genome era, considered this premise, but were limited by the availability of sufficiently long genome sequences, lacked robust direct measurements of repeat instability, and/or considered oversimplified mutational models[3,4,19–33]. Indeed, all such studies of simple repeats have been limited by long-standing technical challenges to sequencing repetitive regions[34–37]. Technological developments led to the release of the human Telomere-to-telomere genome (T2T-CHM13), which more than doubled the number of mapped simple repeats compared to the previous reference genome GRCH38[1]. This warranted a fresh look at the distribution of repeat lengths and whether mutational processes, in the absence of selection, can explain their abundance.

In this study, we measured genome-wide distributions of repeat lengths across mammals, observing that the distribution, including the prevalence of long repeats, is remarkably stable over evolutionary timescales. We modeled the effects of repeat length instability on the evolution of the distribution, finding that the observed repeat length distribution can emerge and be maintained solely due to the interplay between distinct mutational processes. After incorporating empirical estimates and inference of repeat length instability rates, the most parsimonious explanation for the abundance and stability of long repeats does not require invoking selection; rather, extreme mutation rates cause long repeats to emerge as independently evolving elements. We discuss how this collection of observations may inherently constrain mechanistic properties of DNA replication and repair.

## Results
### Features of the repeat length distribution and evolutionary stability

Using T2T-CHM13, we first assembled a genome-wide distribution of repeat tract lengths (henceforth, DRL) for each simple tandem repeat motif, pooling over bioinformatically indistinguishable permutations (see Supplementary Fig. 1a, "Methods" section). Each distribution was assembled by counting contiguous, uninterrupted repetitions of a specified motif, allowing for straightforward bioinformatic assembly of the DRL (see "Methods" section). Each DRL showed a marked excess of repeats longer than ~10 nt, relative to a randomly shuffled control. This was apparent for nearly all motifs (Supplementary Fig. 1b) but with motif-specific variation in the shape of the extended tail of long tract lengths. Figure 1a plots the DRLs after pooling motifs of the same unit length (e.g., mononucleotide repeats, dinucleotide repeats, etc., up to hexamer repeats), each with a clear tail of long repeats. We found that short read sequencing was sufficient to reconstruct the well-populated length classes of nearly all DRLs, lacking estimates only for the very longest repeats (Supplementary Fig. 1c). We were therefore able to estimate distributions from genome sequences of over 300 mammals from the Zoonomia project[38] and compare them to humans. Due to differences in total assembly length, direct comparison was performed on the normalized DRL (see Supplementary Fig. 2, "Methods" section). There was surprisingly little variation in the shape of the DRLs between primates; DRL shapes were qualitatively similar but more variable in mammalian DRLs, consistent with the longer divergence time. This comparison is shown in Fig. 1b for mono-A/mono-T repeats, which are the most prevalent in the human genome and are the primary focus of our subsequent analyses (normalized DRLs for additional motifs shown in Supplementary Fig. 3). Consistency of the shape of the DRLs across the primate lineage suggests that both the repeat tract length distributions and, as a corollary, maintenance of the underlying mechanisms, were largely stable for at least 70 million years. This highly conserved DRL evolution directly suggests the emergence of a steady state equilibrium.

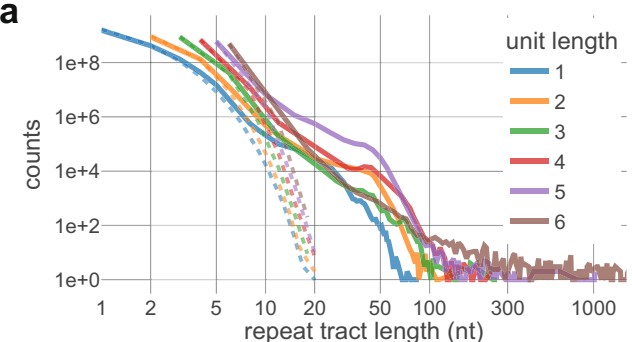

**a**

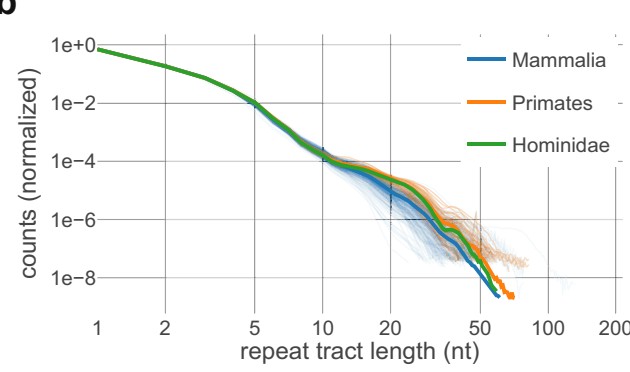

**b**

**Fig. 1 | Distributions of repeat tract lengths (DRLs) by motif length and across phylogenies. a** Counts of repeats in human T2T genome pooled by motif unit length (e.g., unit length 1 pools DRLs for A/T and C/G). Dashed lines represent counts in a randomly shuffled human genome sequence. Canonical centromeric and telomeric motifs are excluded from unit lengths 5 and 6, respectively, due to qualitative differences in the DRLs. **b** Normalized DRLs of mononucleotide-A repeats in mammals (blue; $n = 315$), primates (orange; $n = 37$) and hominids (green; $n = 6$). (See Supplementary Fig. 3 for other motifs.) Counts are necessarily normalized to account for different genome lengths (see "Methods" section, Supplementary Fig. 2). Solid line indicates median values per length bin. Phylogenies are inclusive (e.g., primates are included as a subset of mammals). Thin lines show individual species. Similarity within phylogenies suggests long-term stability of the DRLs.

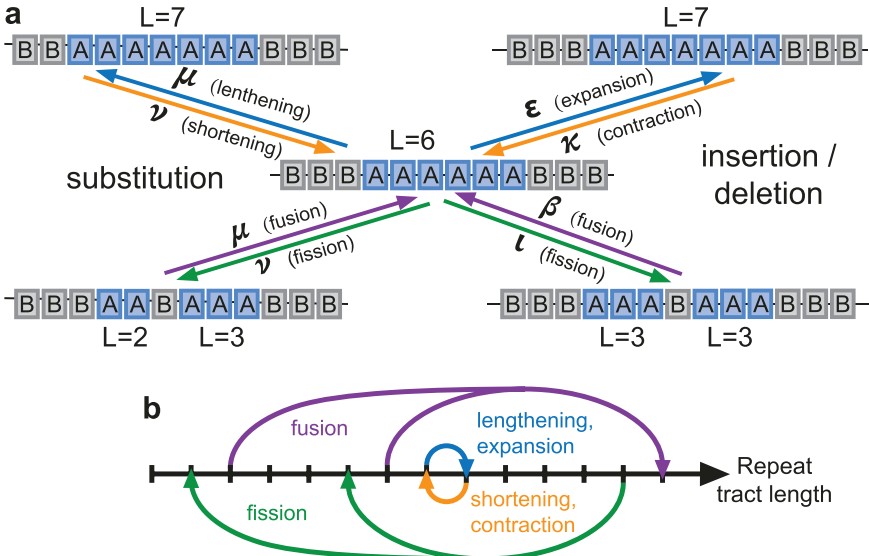

**Fig. 2 | Mutational processes as transitions in repeat tract length. a** Distinct mutational processes, using the example of transitions to and from repeat tract length $L = 6$. 'A' represents a given STR motif; 'B' represents any other sequence with length equal to A. Arrows indicate mutations, either substitutions (left) or indels (right), that affect the length of A repeat tract(s). Mutations can lengthen/shorten (top) or interrupt/rejoin (bottom) repeat tracts. The latter we term repeat 'fission' and 'fusion,' respectively. **b** Depiction of the same mutational processes as length transitions in the DRL. Lengthening/shortening mutations increase or decrease length by one unit ('local' transitions) and maintain the same total count of repeats ('conservative'), while fission/fusion processes are non-local and non-conservative.

The empirical DRLs extend to lengths that, at disease loci, would be subject to genetic anticipation, risking progression to repeat expansion disorders in subsequent generations[5]; despite the associated disease risk, this tail of long repeats appears to be a generic and evolutionarily conserved feature of repeat length distributions. One proposed explanation is that longer repeats confer a selective advantage due to some repeat length-specific biological function[17]. As an alternative, we propose that long repeats emerge and are maintained by the complex interplay between distinct mutational forces. Though these hypotheses are not mutually exclusive, we sought to understand the extent to which mutagenesis, alone, can maintain the shape of the distribution, without introducing natural selection.

## Mutational transitions in repeat tract length

As described above, literature suggests that repeat instability emerges as a very rapid increase in the rate of length changes as tract length increases, with the longest repeats mutating nearly every generation. In light of such high mutation rates, the observation that the DRL evolves in steady state over long timescales is somewhat surprising, suggesting that the maintenance of the distribution results from a dynamic balance between the ensemble of mutational processes that alter repeat length.

To better understand the genome-wide distribution, we therefore require a comprehensive understanding of all involved mutational processes (e.g., nucleotide substitution, insertion, deletion; see Fig. 2 for a schematic of mutational processes) and how they differ by repeat length. Estimating repeat tract lengths from sequencing data is a notorious bioinformatic challenge, particularly for homopolymer repeats[34–37]. Published results only sparsely cover the full range of lengths observed in the genome, largely focusing on disease-relevant lengths and loci[39–45]. In contrast, there is little information about mutation rates at short tract lengths, despite comprising the vast majority of repeats in the genome.

In order to study mutations across a wide range of tract lengths, we first subdivided insertions and deletions into repeat-relevant mutational categories; we refer to expansions and contractions as mutations that alter repeat length by whole motif units and maintain one contiguous repeat tract, in contrast to partial deletions and non-motif insertions. Rates of each mutagenic process were estimated by pooling existing short-read trio sequencing datasets ($n = 9387$ trios; henceforth, 'pooled trio' dataset). This data was sufficient to directly estimate length-dependent rates for short repeats (up to roughly $L = 6-8$ units, depending on motif, where $L$ is the number of repeated units in a tract), but we found that sequencing errors dramatically reduced mutation counts for longer repeat tract lengths (see "Methods" section). We complemented these estimates by length-stratifying data from a recent study[45] that used a population structure-aware caller (named 'popSTR') to study repeat mutations in the mid-to-long length range in short-read trio data ($n = 6084$). Due to a variety of technical considerations (see "Methods" section), estimates were only reliable within a limited range of tract lengths for each motif, which differed by dataset.

It was previously observed that the majority of mutations within a repeat increase or decrease length by one unit (i.e., $L \to L \pm 1$)[46,47]. To expand on this, we length-stratified the mutation data and found that single-unit length changes dominate above a clear length threshold, consistent with the onset of repeat instability (Fig. 3a and Supplementary Fig. 4). Accordingly, we estimated the rates of single-unit length changes, separately estimating the contraction, expansion, and non-motif insertion rates from the pooled trio data; popSTR-based estimates combine expansion and non-motif insertion rates due to technical limitations (see "Methods" section). For mono-A repeats, all instability rates increase rapidly between roughly 5–10 nt (Fig. 3b and see Supplementary Fig. 5 for all motifs), consistent with a threshold-like onset of repeat instability in this length range (detailed below). The popSTR-based estimates suggest that repeat instability rates continue to increase monotonically, at least until the length range where the dataset becomes noisy (Fig. 3b and Supplementary Fig. 5).

The combination of both datasets recapitulates the hallmark of repeat instability[5,6,8,9]: a rapid increase in the rates of expansion and contraction as length increases. Beyond confirming this property, available data were insufficient to robustly estimate the length dependence of each mutational process across the tract length range observed in the genome. Such an estimate is a necessary component of

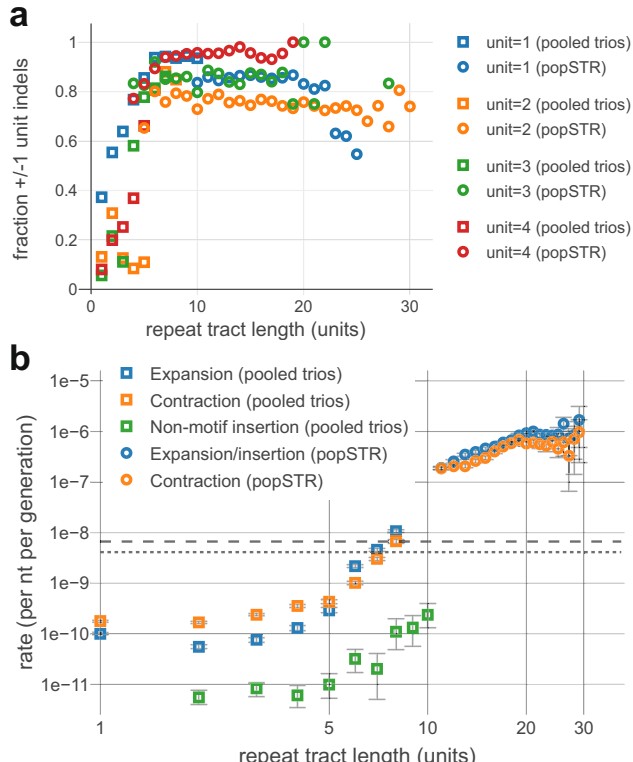

**Fig. 3 | Estimated instability rates stratified by repeat tract length.**
**a** Dependence of indel size on repeat tract length (x-axis) for repeat unit lengths 1–4 (different colors). y-axis measures fraction of all indels that result in single unit length changes (i.e., number of inserted/deleted bases is equal to repeat unit length). Point estimates shown for pooled trio (squares; $n = 9387$) and popSTR datasets (circles; $n = 6084$). Above a threshold of ~5 units, repeat instability primarily consists of ±1 unit changes. Tract lengths subject to severe technical artifacts were omitted for clarity. See Supplementary Fig. 3 for additional detail. **b** Mononucleotide-A mutation rate estimates from pooled trio and popSTR datasets for expansions (blue), contractions (orange), and non-motif insertions (green; note that the popSTR dataset combines expansions and non-motif insertions). Rates calculated only from ±1 unit changes. Point estimates from pooled trio (squares; $n = 9387$) and popSTR datasets (circles; $n = 6084$); statistical error bars represent 95% confidence intervals assuming Poisson mutation counts. Gray dashed and dotted lines show point estimates of substitution rates $\mu$ (A > B, where B = C, G, or T) and $\nu$ (B > A), respectively. Tract lengths subject to severe technical artifacts were omitted (see Supplementary Fig. 4 for complete estimates and additional motifs).

any quantitative understanding of the approach of the DRL towards the steady state (as observed in primates). For further analyses, the length dependence of these mutation rates was extended to longer tract lengths via parameterization in an inference framework described below.

## Computational modeling of DRL dynamics

We sought to assess whether our three observations—the empirical human DRL, the existence of a long-term steady state, and the estimated mutation rates could be simultaneously incorporated into a self-consistent model of repeat length evolution. To this end, we built a computational model that incorporates the length-changing effects of substitutions, expansions, contractions, and non-motif insertions in order to track the evolution of the DRL towards a steady state. We modeled the distribution of mono-A repeats, in part, because they are subject to the simplest ensemble of mutational changes in length (related to the lack of distinction between tract length and the number of repeated units). As a consequence of counting contiguous repeats

to assemble the DRL, mutational processes alter the length of a given repeat in one of four ways (Fig. 2): lengthening, shortening, joining two repeats into one (which we term 'fusion'), or splitting one repeat into two (i.e., repeat interruption, which we term 'fission'). This treatment of interruptions as effectively splitting one repeat into two is consistent with previous observations that interruptions result in locuswide rates that scale with the longest contiguous subunit[48–54] or, equivalently, a rate reduction that scales with distance from the repeat boundary[7].

To reduce the computational time required to evolve a whole genome sequence and simultaneously count contiguous repeat tracts, we directly evolved the DRL by manipulating the occupancy of each length bin. In this formulation, length-altering mutations are reframed as transitions between length bins (see Fig. 2b), and the DRL evolves under repeated application of mutations over many generations. However, the elementary step of the process is deceptively complex, as repeat fusion precludes framing the mutational process as a standard transition rate matrix, and both fission and fusion are nonconservative transitions. We treated the aggregate mutational effects as deterministic, ignoring stochasticity in the mutational process and due to factors like genetic drift, to approximate the expectation of the DRL at late times. We interpret this late-time expectation as an approximation to the steady-state distribution, if one exists, resulting from the modeled mutational processes.

## Bayesian inference and parametric model comparison

We constructed a Bayesian inference procedure (Fig. 4) to constrain properties of repeat instability that are consistent with the steady-state evolution of the observed human DRL (see "Methods" section). The computational model uses explicit length-dependent rates for each mutational process as inputs. We directly incorporated the subset of estimated rates from the pooled trio data shown in Fig. 3b (i.e., for $L = 1–8$); the instability rate curves were then extended to longer lengths to model a rapid, monotonic increase with length. The inclusion of empirical estimates at low lengths, which include a rapid rate increase, limits the number of parameters required to describe complex length-dependent rates of repeat instability with both a rapid transition and a distinct asymptotic functional form for long repeats.

Resemblance to the monotonic increase seen in popSTR estimates at intermediate lengths (Fig. 3b) motivated a class of parameterizations with a power-law increase in the mutation rates (see Table 1, "Methods" section). We first specified a model with minimal degrees of freedom (DoF) that describes equal rates of expansion and contraction (i.e., rates for $L > 8$ represent a simplistic model of replication slippage based on previous literature[8,19,22,27]). This was treated as a null model for comparison to parameterizations with additional DoF that characterize expansion-contraction bias. We used the popSTR-based rate estimates to define plausible, empirically based Bayesian priors for each parameter space, representing varying degrees of confidence in this dataset (Supplementary Fig. 6), including a (naively uninformed) uniform prior. We then used the results of our computationally modeled DRLs in an Approximate Bayesian Computation (ABC) framework (following the prescription in Wilkinson, 2013[55]) to compute a posterior probability distribution for each parametric model. We used the range of primate DRLs to define a rejection probability by comparing them to the human DRL (using Kullback-Leibler (KL) divergence to quantify the difference between two distributions). The same quantity was computed for each computationally modeled DRL and used to approximate the posterior probability distribution over the parameter space (see "Methods" section for details).

The results of our inference for each parameterization are summarized in Table 1. We assessed the relative statistical support for various model comparisons via the Bayes factor ratio (see "Methods" section). The relative Bayes factors strongly suggested discarding the

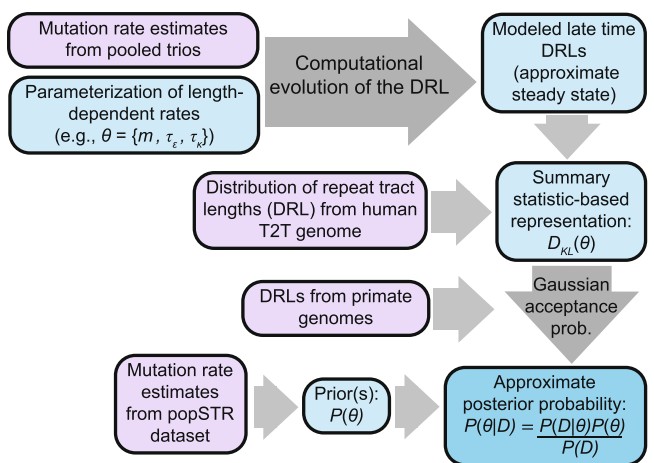

**Fig. 4 | Schematic representation of the Bayesian inference procedure.** Inference of parameters representing length-dependent instability rates via Approximate Bayesian Computation (ABC). Empirical data sources informing the inference are shown in purple. First, expansion and contraction rates are parameterized at lengths where estimates are unreliable or unavailable. Each parameter combination specifies a complete set of mutation rates for repeats of all lengths. Given an initial state, mutational transitions (see Fig. 2) are repeatedly applied to evolve the DRL for a large number of generations. The late-time DRL is treated as an approximation to the steady-state DRL (when applicable). For each parameter combination, the difference between the late-time DRL and the human T2T DRL is summarized by the Kullback−Leibler (KL) divergence. The approximate posterior is proportional to the product of the prior and the probability of acceptance under an ABC rejection strategy[55]. The prior is informed by popSTR-estimated mutation rates. The acceptance probability is treated as Gaussian-distributed in the KL divergence, with mean zero and variance defined by divergences for an ensemble of primates (see "Methods" section). This quantity is calculated for each parameter combination and subsequently normalized over parameter space to approximate the posterior probability distribution.

two-parameter null model in favor of further DoF that introduce asymmetry (i.e., bias) between the expansion and contraction rates. The power-law parameterization with the largest Bayes factor, regardless of prior, was a three-dimensional model of expansion and contraction with distinct exponents and related multiplicative constants (see Table 1, "Methods" section). This model is a modest improvement over the four-parameter description (i.e., completely decoupled expansion and contraction rates), which otherwise provides a dramatically better description than lower-dimensional parameterizations.

To test how reliant these conclusions are on the power-law functional form, we defined an alternate class of parameterizations with logarithm-based growth rates (i.e., with slower-growing rates at large lengths to better approximate saturation). Model comparison within this class of parameterizations provided qualitatively consistent results to those comparing power law parameterizations (see Table 1). The inequivalence of priors across functional forms (along with additional necessary approximations; see "Methods" section) suggests caution should be taken in direct comparisons between models with distinct functional forms. Due to the relative analytic simplicity of the functional form, subsequent analyses were focused on power-law parameterization results.

### Inference of instability rates from the steady-state repeat length distribution

Amongst power-law models, we focused on the three-parameter multiplier-coupled model, as it showed the strongest statistical support, regardless of the choice of prior. Above $L = 8$, this model is parameterized by exponents, $\tau_\epsilon$. and $\tau_\kappa$, and a common multiplier $m$

(representing a discrete jump in rates immediately above the empirical estimates; explicit definition in "Methods" section), which together characterize the length dependence of expansion $\epsilon(L; m, \tau_\epsilon)$ and contraction $\kappa(L; m, \tau_\kappa)$ (see "Methods" section, Table 1 for full definitions). The common multiplier for expansion and contraction, which limits the dimensionality, may be interpreted as representing the onset of repeat instability due to some common biological mechanism. Our trio rate estimates rapidly rise in the length range where they lose accuracy; $m$, which describes a potentially dramatic jump immediately above this range, can provide an oversimplified characterization of a rapid transition to power-law-like behavior. To limit further DoF, we assumed that the length dependence of non-motif insertions is dictated by $\tau_\epsilon$, the expansion rate exponent, due to their parallel increase in de novo rates (Fig. 3b) and because they likely arise from the same biological mechanism (e.g., synthesis of the inserted nucleotides by an error-prone polymerase). The parameter space we explored includes the possibility of a constant per-nucleotide rate (i.e., $\tau = 0$, analogous to the constant per-nucleotide substitution rates), linearity (i.e., $\tau = 1$), a natural conceptual model for length dependence associated with repeat instability, and more rapid growth on par with popSTR-based estimates (Fig. 3b). However, the parameterization itself is not intended to represent a specific biological model; the true rate curves are likely more complex due to multiple contributing mechanisms.

To interpret the resulting posteriors, we first approximated highest density regions (HDRs) comprising 68%, 95%, and 99.7% of posterior probability on the finite grid for the multiplier-coupled model (Fig. 5a and Supplementary Fig. 7a; alternate parameterizations shown in Supplementary Figs. 8 and 9). The posterior is largely localized along a ridge of constant values of $\Delta\tau \equiv \tau_\kappa - \tau_\epsilon$ (roughly, $\Delta\tau \approx 0.3{-}0.6$) and roughly between multipliers $m = 1.6{-}6.4$; the difference between expansion and contraction rate exponents appears to be more relevant than their specific values. Under the uniform prior, some parameter combinations within the 95% HDR deviate from this range of $\Delta\tau$ (extending to both lower values of $\Delta\tau$ and lower $\tau_\epsilon$, $\tau_\kappa$; Fig. 5a) but are excluded when applying the popSTR-based prior, which requires consistency with the larger estimated instability rates (Fig. 5a and Supplementary Fig. 7a).

We computed the posterior-weighted DRL (i.e., the expectation value of the DRL; see "Methods" section) and the range of DRLs consistent with the 95% HDR (Fig. 5b). The posterior-weighted DRLs closely resemble the human genome-wide distribution, while the 95% HDR parameters roughly span the range of primate DRLs used in our inference procedure. This demonstrates that the coarse features of the empirical DRL can be recapitulated from mutational dynamics alone.

We then computed the posterior-weighted length-dependent rates of expansion and contraction for each prior and found rough consistency with popSTR-estimated rates (Fig. 5c). One salient feature emerged, regardless of prior: expansion bias at intermediate tract lengths transitions to contraction bias at longer lengths due to the faster increase in contraction rate with length (i.e., $\tau_\kappa > \tau_\epsilon$; see Fig. 5a, c). This likely explains the preference for the multiplier-coupled model, which necessarily inherits a modest initial expansion bias directly from empirical rate estimates. However, if the apparent expansion bias at $L = 8$ is simply a consequence of homopolymer sequencing errors, correcting the direction of this bias would lead to statistical rejection of the multiplier-coupled model in favor of the four-parameter model (which lacks the a priori imposition of initial expansion bias on the parametrized rates). Regardless, inference results under the four-parameter model recapitulate the importance of a transition from expansion to contraction bias (Supplementary Fig. 10).

To gain intuition for the preference of expansion-to-contraction biased parameters in the multiplier-coupled model, we contrasted the DRLs to the parameter combinations outside of the 95% HDR. Excluding slowly evolving rates, the remaining parameter space

**Table 1 | Repeat instability rate parameterizations and Bayesian inference results**

| Model name | Parameters | Functional form | Prior | Bayes factor (ratio to null) | Max posterior [mean posterior] |
|---|---|---|---|---|---|
| *Symmetric power law*[†] (L < 9 from empirical rates) | 2; $\theta = (c, \tau)$ | $\epsilon(L>8) = \kappa(L>8) = c\left(\frac{L}{9}\right)^\tau$ | Uniform | 5.1e-18 (1.0) | $\theta = (4.3\text{e-}8, 0.6)$ [[4.3e-8, 0.67]] |
| | | | Permissive | 2.8e-21 (1.0) | $\theta = (4.3\text{e-}8, 1.2)$ [[4.1e-8, 1.3]] |
| | | | Restrictive | 7.6e-31 (1.0) | $\theta = (2.7\text{e-}8, 3.0)$ [[3.4e-8, 2.8]] |
| *Decoupled power laws* (L < 9 empirical) | 4; $\theta = (c_\epsilon, c_\kappa, \tau_\epsilon, \tau_\kappa)$ | $\epsilon(L>8) = c_\epsilon\left(\frac{L}{9}\right)^{\tau_\epsilon}$ $\kappa(L>8) = c_\kappa\left(\frac{L}{9}\right)^{\tau_\kappa}$ | Uniform | 2.0e-5 (3.9e12) | $\theta = (2.7\text{e-}8, 1.7\text{e-}8, 1.6, 2.0)$ [[2.9e-8, 1.8e-8, 1.9, 2.3]] |
| | | | Permissive | 1.9e-5 (6.9e15) | $\theta = (2.7\text{e-}8, 1.7\text{e-}8, 3.0, 3.5)$ [[3.9e-8, 2.5e-8, 2.9, 3.3]] |
| | | | Restrictive | 2.2e-5 (2.9e25) | $\theta = (4.3\text{e-}8, 2.7\text{e-}8, 3.1, 3.6)$ [[4.7e-8, 2.9e-8, 3.1, 3.6]] |
| *Power laws with independent constants* (L < 9 empirical) | 3; $\theta = (c_\epsilon, c_\kappa, \tau)$ | $\epsilon(L>8) = c_\epsilon\left(\frac{L}{9}\right)^{\tau}$ $\kappa(L>8) = c_\kappa\left(\frac{L}{9}\right)^{\tau}$ | Uniform | 1.3e-5 (2.6e12) | $\theta = (1.7\text{e-}8, 1.1\text{e-}8, 0.9)$ [[2.0e-8, 1.2e-8, 0.6]] |
| | | | Permissive | 9.9e-14 (3.5e7) | $\theta = (1.7\text{e-}8, 1.1\text{e-}8, 1.0)$ [[1.7e-8, 1.1e-8, 1.0]] |
| | | | Restrictive | 2.8e-31 (3.7e-1) | $\theta = (2.7\text{e-}8, 2.7\text{e-}8, 3.0)$ [[3.4e-8, 3.4e-8, 2.8]] |
| *Power laws with independent exponents* (L < 9 empirical) | 3; $\theta = (c, \tau_\epsilon, \tau_\kappa)$ | $\epsilon(L>8) = c\left(\frac{L}{9}\right)^{\tau_\epsilon}$ $\kappa(L>8) = c\left(\frac{L}{9}\right)^{\tau_\kappa}$ | Uniform | 9.7e-7 (1.9e11) | $\theta = (2.7\text{e-}8, 0.6, 0.1)$ [[2.7e-8, 2.3, 2.1]] |
| | | | Permissive | 2.3e-6 (8.2e14) | $\theta = (2.7\text{e-}8, 3.4, 3.3)$ [[2.8e-8, 3.3, 3.2]] |
| | | | Restrictive | 2.0e-6 (2.6e24) | $\theta = (2.7\text{e-}8, 3.6, 3.5)$ [[2.9e-8, 3.5, 3.4]] |
| **Multiplier-coupled power laws (L < 9 empirical)** | 3; $\theta = (m, \tau_\epsilon, \tau_\kappa)$ | $\epsilon(L>8) = \epsilon(8) \times m\left(\frac{L}{9}\right)^{\tau_\epsilon}$ $\kappa(L>8) = \kappa(8) \times m\left(\frac{L}{9}\right)^{\tau_\kappa}$ ($\epsilon(8)$, $\kappa(8)$ from empirical estimates) | **Uniform** | **1.8e-4 (3.5e13)** | $\theta = (2.5, 1.6, 2.0)$ [[2.7, 1.9, 2.3]] |
| | | | **Permissive** | **1.4e-4 (4.8e16)** | $\theta = (4.0, 2.8, 3.3)$ [[3.6, 2.9, 3.3]] |
| | | | **Restrictive** | **1.3e-4 (1.7e26)** | $\theta = (4.0, 3.1, 3.6)$ [[4.3, 3.1, 3.6]] |
| *Symmetric logarithmic power*[††] (L < 9 from empirical rates) | 2; $\theta = (c, \tau)$ | $\epsilon(L>8) = \kappa(L>8) = c\left(\frac{\log(L-7)}{\log 2}\right)^{\tau}$ | Uniform | 8.4e-18 (1.0) | $\theta = (2.7\text{e-}8, 0.8)$ [[3.2e-8, 0.7]] |
| | | | Permissive | 3.4e-19 (1.0) | $\theta = (2.7\text{e-}8, 0.9)$ [[3.4e-8, 0.7]] |
| | | | Restrictive | 8.3e-29 (1.0) | $\theta = (4.3\text{e-}8, 1.0)$ [[4.2e-8, 1.0]] |
| *Decoupled logarithmic power* (L < 9 empirical) | 4; $\theta = (c_\epsilon, c_\kappa, \tau_\epsilon, \tau_\kappa)$ | $\epsilon(L>8) = c_\epsilon\left(\frac{\log(L-7)}{\log 2}\right)^{\tau_\epsilon}$ $\kappa(L>8) = c_\kappa\left(\frac{\log(L-7)}{\log 2}\right)^{\tau_\kappa}$ | Uniform | 1.8e-4 (2.1e13) | $\theta = (2.7\text{e-}8, 1.1\text{e-}8, 2.1, 2.8)$ [[3.4e-8, 1.4e-8, 1.7, 2.4]] |
| | | | Permissive | 5.1e-4 (1.5e15) | $\theta = (4.3\text{e-}8, 1.7\text{e-}8, 2.0, 2.7)$ [[3.8e-8, 1.4e-8, 2.1, 2.9]] |
| | | | Restrictive | 1.8e-3 (2.2e25) | $\theta = (4.3\text{e-}8, 1.7\text{e-}8, 2.0, 2.7)$ [[3.8e-8, 1.3e-8, 2.2, 3.1]] |
| *Multiplier-coupled logarithmic power* (L < 9 empirical) | 3; $\theta = (m, \tau_\epsilon, \tau_\kappa)$ | $\epsilon(L>8) = \epsilon(8)m\left(\frac{\log(L-7)}{\log 2}\right)^{\tau_\epsilon}$ $\kappa(L>8) = \kappa(8)m\left(\frac{\log(L-7)}{\log 2}\right)^{\tau_\kappa}$ ($\epsilon(8)$, $\kappa(8)$ from empirical estimates) | Uniform | 3.9e-4 (4.6e13) | $\theta = (2.5, 1.3, 0.9)$ [[2.8, 1.0, 1.2]] |
| | | | Permissive | 2.7e-4 (7.7e14) | $\theta = (4.0, 1.4, 1.7)$ [[3.5, 1.5, 1.8]] |
| | | | Restrictive | 1.7e-4 (2.0e24) | $\theta = (4.0, 1.8, 2.1)$ [[4.0, 1.9, 2.2]] |
| *Pure power law* (parameterized at all lengths) | 4; $\theta = (\lambda_\epsilon, \lambda_\kappa, \tau_\epsilon, \tau_\kappa)$ | $\epsilon(L) = \mu\left(\frac{L}{\lambda_\epsilon}\right)^{\tau_\epsilon}$ $\kappa(L) = \nu\left(\frac{L}{\lambda_\kappa}\right)^{\tau_\kappa}$ (empirically estimated sub. rates $\mu$, $\nu$) | Uniform | 3.1e-17 | $\theta = (9, 13, 3.6, 4.0)$ [[9.1, 12.9, 3.7, 4.0]] |
| | | | Restrictive | 8.7e-17 | $\theta = (9, 12, 3.8, 4.0)$ [[9.0, 12.5, 3.7, 4.0]] |

null model for [†]power-law and [††]logarithmic power models; **bold**: largest Bayes factor amongst power-law models

*Parametric models of instability rates and summary of Bayesian inference results*. For each parameterization used in our analyses, this table specifies the model name (as referred to in the text), the tract lengths described by the parameterization, the inference parameters, the functional forms for length-dependent expansion and contraction rates, and a summary of inference results. For each model, the following quantities are given for each prior: Bayes factor (and Bayes factor ratio to null model within the same nesting, denoted by symbols), parameter combination with maximum posterior probability, and mean posterior parameter combination. The primary model considered is shown in bold text. Further details on prior construction, calculation of Bayes factors (and model comparison), and expectation used to compute mean posterior parameters are provided in the "Methods" section.

broadly separates into three qualitative categories, largely characterized by $\Delta\tau$: $\Delta\tau \gtrsim 0.6$ yields DRLs that underestimate the long repeat tail (i.e., early truncation), while $0 < \Delta\tau \lesssim 0.2$ yields distributions that overestimate the long tail (Supplementary Fig. 11). The roughly half of parameter space with $\Delta\tau < 0$ showed a clear reason for near-zero posterior probabilities: regardless of multiplier, these parameter combinations do not converge to steady state at late times and are subject to explosive growth in all length bins (Supplementary Figs. 11 and 12a).

In addition to $\Delta\tau$ values outside of the high posterior ridge, larger values of $m$ generally remained beyond the 95% HDR. This was likely due, in part, to the discrete jump between estimated and

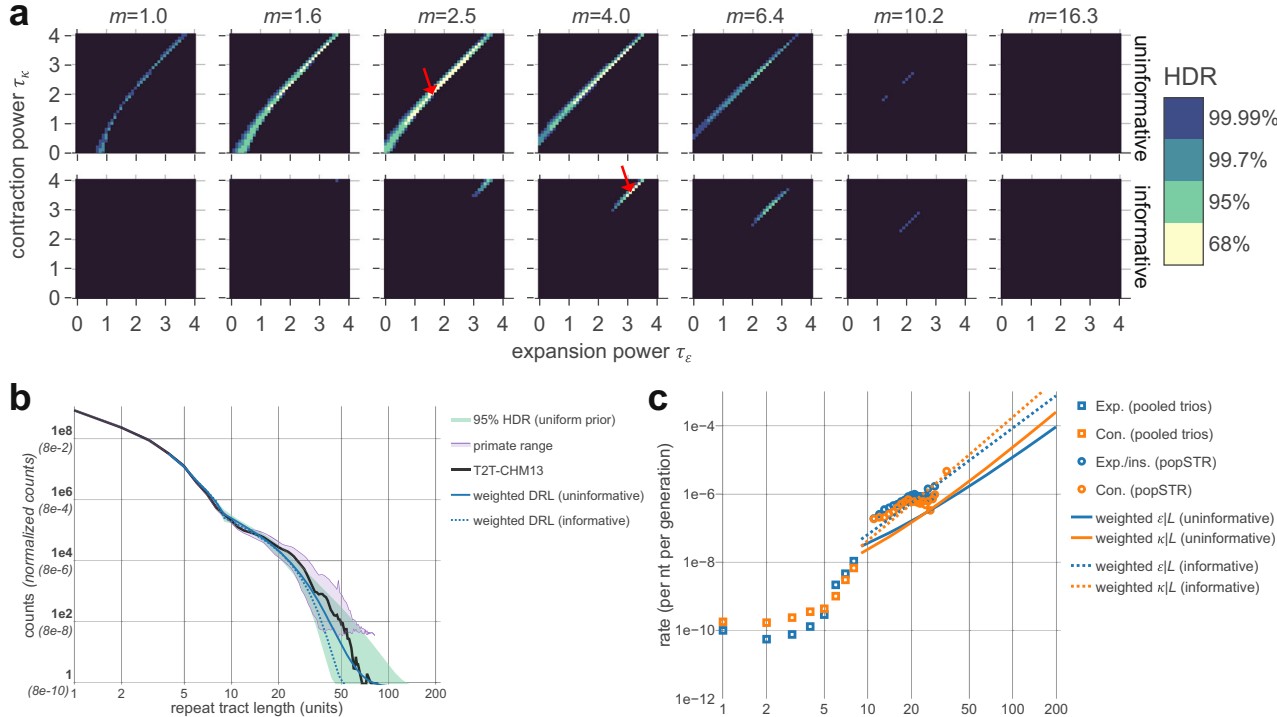

**Fig. 5 | Inference results and self-consistency of a mutation-only model. a** Bayesian posterior probabilities inferred for the three-parameter multiplier-based power-law model of mono-A repeat instability rates; uninformative prior (top row), popSTR-based informative prior (bottom row; restrictive condition, see Supplementary Fig. 5). Each coordinate represents a distinct set of length-dependent rates defined by parameters $(m, \tau_\epsilon, \tau_\kappa)$; $\tau_\epsilon$ (x-axes) and $\tau_\kappa$ (y-axes) determine the power laws for expansion and contraction, respectively, and $m$ (columns) represents a multiplicative jump at $L = 9$ (parameterization in Table 1). Color indicates the highest density range (HDR) of the posterior for various total probabilities; the black region sums to 0.01% of the probability. Red arrows show the maximum posterior for each prior. Informative prior results in a more rapid increase in instability rates with length. Supplementary Figs. 6–9 show posteriors under various parameterizations. **b** Comparison of inference to empirical DRLs. DRLs are necessarily normalized for comparison (conditional on $L > 3$, see "Methods" section); the y-axis indicates normalized fractions (parentheses) and counts rescaled to match the number of repeats in the T2T genome (bold labels). Blue lines represent posterior-weighted DRLs (average of all DRLs weighted by the posterior

probability for each parameter combination; see "Methods" section) for informative (dashed) and uninformative (solid) priors; modeled DRLs are largely consistent with the empirical T2T DRL (black). The green region shows the minimum and maximum counts at each length bin across all parameters within the 95% HDR (uninformative prior). Purple region shows min-max range generated from non-human primate genomes ($n = 34$, after removing the two most-diverged DRLs and truncating each DRL where raw counts drop below 30; see "Methods" section). Overlap between these regions indicates that the posterior under the uninformative prior largely reflects the ensemble of primates. **c** Posterior-weighted repeat instability rates. Tract length dependencies of expansion (blue) and contraction rates (orange) for uninformative (solid) and informative (dashed) priors. Empirical estimates from pooled trios (squares; directly incorporated in the model) and popSTR data (circles; used to construct informative priors) are shown for comparison. Informative prior imposes consistency with popSTR-estimated rates, while the posterior-weighted DRL (**b**) remains consistent with the T2T genome.

parameterized rates at L = 9 that results in a discontinuous DRL, which can artificially inflate the KL divergence. To investigate this, we repeated our inference after smoothing the instability rate length dependences via interpolation around this transition (Supplementary Fig. 8). The results suggested that a (naively more realistic) interpolated length dependence results in less penalization for larger multipliers and posterior probabilities more robust to the choice of prior. Additionally, inference using interpolated rates under the restrictive informative prior results in posterior-weighted instability rate estimates that overlap the popSTR estimates (Supplementary Fig. 8c); this suggests a smoother length dependence may represent more realistic instability rates. Indeed, this mutational model incorporates all available data and describes a self-consistent picture of a steady-state DRL shaped only by mutational dynamics.

To better understand the approach to steady state for realistic parameters within the 95% HDR, we followed the temporal evolution of the DRL, starting from a highly diverged initial state (see "Methods" section, Supplementary Fig. 12b). This analysis suggested a two-stage equilibration process with two distinct timescales. The bulk of the long repeat tail establishes exponentially quickly, followed by a slower fine-scale equilibration of mutational processes at each length. Finally, we

tested for robustness to potential confounders (e.g., differing initial conditions, use of a step-wise speed-up factor, lack of stochastic fluctuations, etc.) and found no major changes in the qualitative results (see "Methods" section, Supplementary Fig. 13). Collectively, these results show that mutational dynamics, rather than natural selection, may be responsible for the maintenance of an excess of mid-to-long tract length repeats in the human genome.

## Maintenance of the repeat length distribution in steady state

To understand the complex interplay between mutational processes that shapes and stabilizes the distribution of repeat lengths, we constructed an analytic model of the dynamics. This analytic approximation captures the behavior of the DRL after the mutational process reaches steady state (see "Methods" and "Supplementary Note"), focusing primarily on the previously described multiplier-coupled three-dimensional parameterization. A number of previous studies have constructed mathematical models of repeat instability to study repeat length evolution[19,22,25–29,31,33,56], including a notable study by Lai and Sun[30] that incorporates many of the elements detailed herein. However, the combination of empirical rate estimates, a robust genome assembly, and our phylogenetic observations motivated the

construction of a model from first principles that is directly informed by this collection of observations. In addition to differences in mathematical machinery, the analytic construction differs from previous efforts by incorporating pervasive length-dependent expansion-contraction bias (Fig. 3b and Supplementary Fig. 5) and explicit effects from non-motif insertions.

We first constructed a discrete equation for the change in the number of repeats at a given length in a single generation due to the deterministic action of mutations (i.e., in the absence of selection and stochasticity in the mutational process, consistent with our computational model). We then imposed a steady state condition by requiring that the sum of all changes in and out of each length class vanishes at each time step after equilibration. Despite the simplifying assumption of steady state, the full dynamical equation cannot be solved generically. However, our estimates of de novo mutation rates suggested a dichotomy exists in the primary driver of changes in length between short and longer repeats (i.e., primarily substitutions for $L < 8$ A-mononucleotide repeats vs expansions and contractions for $L > 10$; see below for direct inference of this length range). Accordingly, short and long repeat dynamics can be treated as separable (i.e., under the approximation of a separation of repeat length scales), leading to simpler approximations of both length regimes. Transitions between the short and long repeat regimes, while present, remain negligible in all realistic scenarios (see "Supplementary Note").

For short repeats, we treated indel mutations as negligible and showed that a geometric distribution (see Methods, Eq. 8) exactly solves the steady state equation under two-way substitutions alone (see "Supplementary Note"). For longer repeats, we constructed a partial differential equation (PDE) that approximates the discrete equation and studied its time-independent properties in steady state; dynamical equations are derived in the "Supplementary Note" in terms of generic parameterization of the length-dependent instability rates. Focusing on the multiplier-coupled parameterization, we obtained numerical solutions to the steady-state dynamical equations under various approximations and under the assumption that fusion-based contributions are negligible to long repeats (see "Methods" section, Eqs. 9−11 and Supplementary Note). These solutions, along with the geometric distribution for short repeats, accurately describe the late-time DRLs produced by our computational model across the range of parameters that approach a steady state (Fig. 6, Supplementary Figs. 14−17, and Supplementary Note). Using these comparisons, we found that, within some parameter regimes, the dynamics simplify to a less complex balance of mutational processes (see "Methods" section, Eqs. 10−11 and Supplementary note) and assessed the appropriate regime of validity (Fig. 6a, b and Supplementary Figs. 14−17). To more directly test the accuracy of the PDE, we used our computational results to decompose the per-generation fluxes in and out of each length class into relative contributions from each mutational type. This allowed for identification of the dominant mutational processes maintaining steady state (Fig. 6c and Supplementary Note); the accuracy of each approximation was confirmed by analyzing the net magnitudes of fission and fusion within each length class and regime (Supplementary Figs. 18−20).

We used this model to study the shape and stability of the empirical DRL and distinctions between repeats in different length regimes under mutational forces alone. Expansions and contractions remain non-negligible for any long repeat across the space of parameters that lead to stable late-time DRLs, highlighting the importance of repeat length instability to the maintenance of long repeats. For extreme parameters that stabilize (i.e., $\tau_\kappa \gg \tau_\epsilon$), the dynamics of all long repeats are dominated by expansion and contraction, alone, leading to a DRL that truncates more rapidly than under substitutions alone (i.e., a *depletion* of long repeats relative to a geometric distribution). In contrast, for realistic parameters (i.e., within the 95% HDR for A-mononucleotide repeats), an intermediate length regime emerges,

characterized by the relevance of repeat fission. An accurate description of the shape of the DRL requires fission to account for the loss of repeats from the extreme tail (i.e., the longest populated length bins) and gain of intermediate length repeats. The relative contributions of fission due to substitutions and non-motif insertions are parameter-dependent; within the rough neighborhood of the maximum posterior parameters (informative prior), substitution is the primary driver of fission up to lengths of ~20 nt, while longer repeats are primarily interrupted by non-motif insertions (see "Supplementary Note"). Fission-based losses in the extreme tail are insufficient to fully counteract length increases due to expansion, independent of the mutational mechanism and parameter values. Instead, contraction is primarily responsible for truncating the DRL at finite repeat length but can be bolstered by both substitution- and non-motif insertion-based fission. The dynamics of the long repeat regime decouples from that of short repeats such that rapidly mutating long repeats effectively become independently evolving genomic elements, categorically distinct from random sequences of the same length. The abundance of long repeats in the genome may therefore be a consequence of their largely unencumbered evolution caused by rapid changes in length.

## Inferring the onset of instability from the shape of the DRL

We sought to better characterize the length at which repeats become independently evolving genomic elements. Our analyses thus far suggest that this occurs roughly at the length where expansion and contraction rates exceed substitution rates. This length was explicitly fixed in our inference via reliance on empirical rate estimates at $L = 1−8$; however, this precluded exploration of the onset length of repeat instability. To study the encoding of this information within the shape of the DRL, we defined fully parameterized rate curves (omitting all empirical rate estimates) that include the length at which instability rates exceed substitution as an explicit parameter. Expansion and contraction rates are each parameterized by an independent power law at all lengths (i.e., with no reference to empirical estimates) in terms of an exponent $\tau$ and $\lambda$, the length at which the rate intersects the relevant substitution rate ($\mu$ for expansion, $\nu$ for contraction; full parameterizations defined in Table 1, see "Methods" section). This four-parameter model depicts an oversimplified rate dependence but serves as a toy model to probe the instability onset length $\lambda$.

Applying the same Bayesian inference pipeline, we estimated the posterior probability using both a uniform prior (i.e., excluding all rate data) and a prior informed by the combined set of empirical rate estimates (i.e., from both the pooled trio and popSTR data; see "Methods" section). We marginalized the posterior to specify values of the onset lengths for expansion ($\lambda_\epsilon$) and contraction ($\lambda_\kappa$) and found highly restrictive marginal distributions with 95% HDRs isolated to $\lambda_\epsilon = 9$, $\lambda_\kappa = 12−13$ (informative prior: $\lambda_\epsilon$, $\lambda_\kappa = 9$, 12; Supplementary Fig. S21 and Table 1). This recapitulates the range of lengths observed in direct empirical rate estimates, despite excluding all such data from the inference (i.e., isolating the influence of the shape of the DRL). The posterior-weighted DRLs reproduce the informative features of the human DRL (e.g., deviation from the substitution-driven geometric distribution at roughly 10 nt; Supplementary Fig. 21b), despite the oversimplified model of instability. This suggests that the transition in shape of the DRL corresponds to the onset length of repeat instability, allowing for rough estimation of this key feature from visual inspection of the distribution.

## Application to repeats with longer unit length

Given that the DRL is informative about the onset length of repeat instability, we next compared this quantity across motifs of differing unit lengths (e.g., dinucleotides, trinucleotides). Empirical rate estimates for all motifs showed qualitatively similar properties to mono-A repeats (i.e., predominantly single-unit expansions and contractions with rates that scale rapidly with tract length; Fig. 3a and

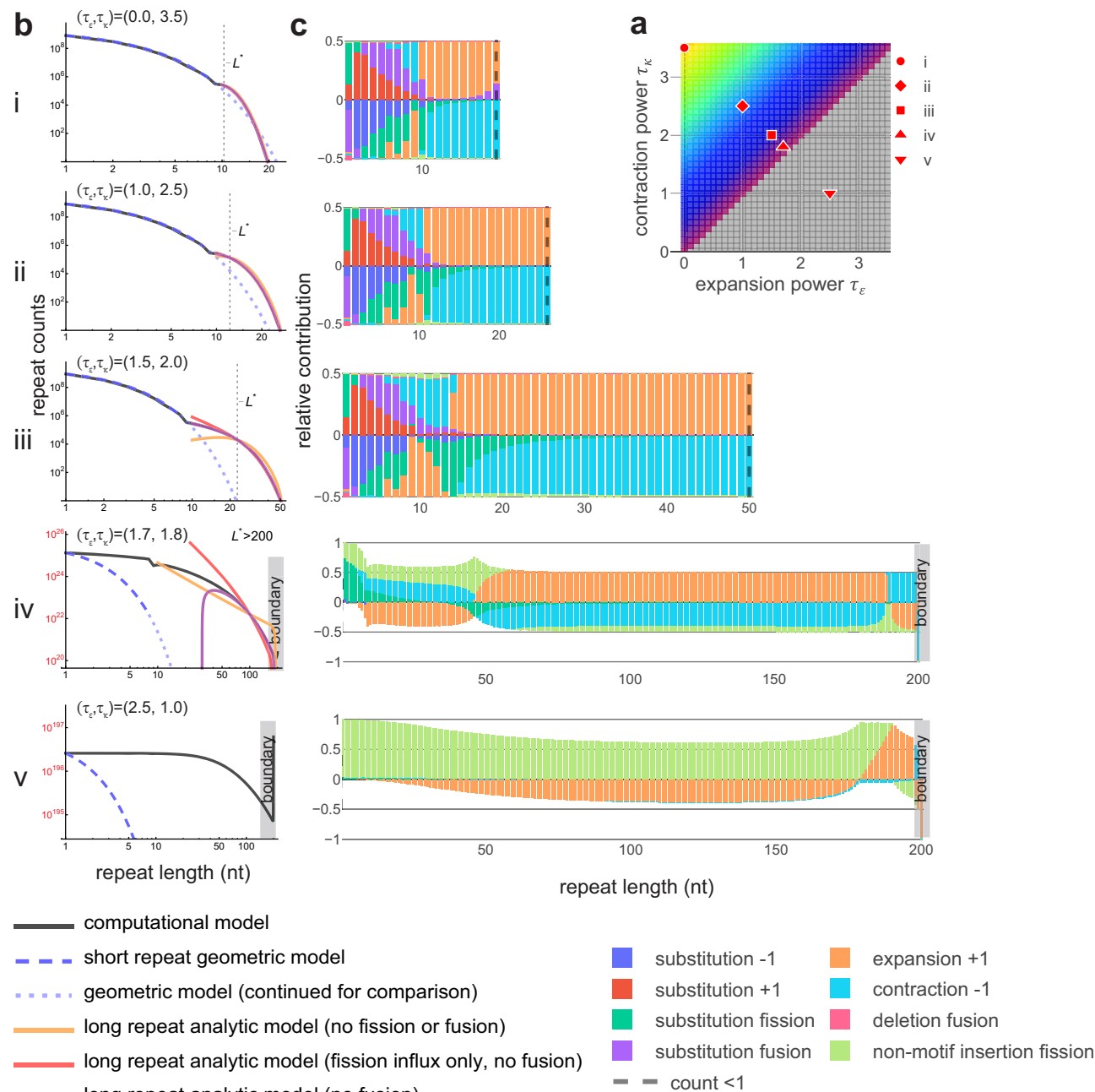

**Fig. 6 | Dynamical regimes distinguishable by dominant mutational effects. a** Slice of parameter space for the three-parameter multiplier-based model (parameterization in Table 1). Five example parameter combinations with $m = 4$ and $\tau_\varepsilon + \tau_\kappa = 3.5$ are shown, corresponding to plots in (**b**, **c**). Colors roughly divide the parameter space into dynamical regimes. **b** Comparisons between computational model results and numerical solutions of approximate steady state equations (see "Methods" section, Supplementary Note). The short length regime at equilibrium is geometrically distributed (blue dashed lines). For long repeats, numerical solutions are shown for three nested approximations to the steady state equation in the continuum limit ($L \gg 1$) in the absence of fusion (due to negligible rates); solutions to approximations in Eqs. 10–12 are shown in orange, red, and purple, respectively. Local transitions ($L$ to $L \pm 1$) were modeled as a combination of symmetric (diffusive) and asymmetric (directional bias) components. $L^*$ represents the length at which expansion and contraction rates are equal. For strong asymptotic contraction-bias (**b**: i), all three approximations remain valid, indicating that the dynamics are well approximated by neglecting fission entirely (orange curve

aligns). For moderate contraction bias (**b**: ii), outflux due to fission becomes non-negligible (orange begins to deviate). For realistic parameter combinations with lower contraction bias (**b**: iii), outflux due to fission is required at all lengths (orange fails); influx due to fission is required at intermediate lengths (red deviates); fusion remains negligible (purple remains accurate). Plots (**b**: iv–v) display non-equilibrium dynamics leading to a rapid increase in repeat counts and explosive growth in genome size. Under universal expansion bias (**b**: v), DRL extends indefinitely above the boundary imposed at $L = 200$ (for computational feasibility). Steady-state analytics do not apply. **c** Relative contributions from each mutational transition to net flux (in minus out) per length bin, produced by computational model. Dashed line indicates DRL truncation (counts < 1). Consistent with analytic predictions, fission is subdominant under strong contraction bias, has relevant outflux under moderate to weak contraction bias, and relevant influx at intermediate lengths under weak contraction bias. Equilibrium distribution is stabilized in detailed balance (net influx = outflux). Influx > outflux (**c**: iv–v) leads to non-equilibrium dynamics and indefinite genome growth.

Supplementary Figs. 4 and 5), suggesting an analogous dynamical competition between substitutions and repeat instability that shapes the steady state DRL. We compared two distinct measures of the onset of repeat instability, representing long- and short-timescale information: first, rough lengths at which empirical DRLs first deviate from geometric decay (i.e., the expected DRL under substitutions alone) and second, the approximate lengths at which per-repeat expansion and/or contraction rates first exceed either substitution rate (i.e., rates comparable to $\mu$ or $\nu$ that perturb the geometric dependence on $L$). Both measures showed reasonable agreement confined to a range of onset lengths between roughly 6 and 12 nt, despite differing unit lengths (Supplementary Fig. 22a). This suggests a universal description of repeat dynamics that shapes the extended tail of the DRL, despite apparent differences in the geometric portion at short lengths. Rapid geometric falloff is an immediate consequence of increasing the unit length (Supplementary Fig. 22b): while a single substitution is sufficient to shorten tract lengths regardless of motif, lengthening of repeat tracts can require multiple substitutions (up to the unit length of the motif). Given initially comparable expansion and contraction rates across motifs, longer motifs show a more immediate transition to the repeat instability-dominated dynamics when measured in the number of repeated units but largely agree when measuring the onset length of repeat instability in nucleotides (Fig. 1a).

## Discussion

Motivated to understand the origin, prevalence, and maintenance of simple tandem repeats in the genome, we constructed a model of repeat evolution under mutagenesis alone that bridges short- and long-timescale observations of repeat length instability. We demonstrate that mutations alone are sufficient to explain the shape of the genome-wide distribution of tract lengths. The abundance of long repeats in the genome reflects the rapid onset of repeat instability with an initial expansion bias, rather than natural selection. This observation does not preclude selection at specific loci, whether beneficial or disease-associated, provided these comprise a small portion of repeats in the genome.

Length-dependent expansion-contraction bias is evident in our de novo estimates; incorporating this property into the mutational model is sufficient to truncate the distribution at finite lengths due to substantial contraction-bias. The long length tail of the distribution is produced and maintained in a dynamic balance between expansion, contraction, and fission. This implicitly prevents the growth of repeats to disease-relevant lengths, suggesting natural selection as a disease-prevention mechanism may not be essential. If selection, rather than contraction bias, is responsible for terminating the distribution below disease length, it would have to be enormously efficient to counteract instability-driven expansion rates and act globally across all sufficiently long repeats. If pervasive selection plays a role in shaping the distribution, this must be inferred as a deviation from the DRL under mutation alone, built on a more complete model of repeat mutagenesis.

Our analysis of the genome-wide properties of repeats is complementary to studies of individual loci harboring disease, which generally occurs at or above the longest lengths present in the reference genome (i.e., stochastically driven length classes in the present context). Such elongated repeats can form motif-specific secondary structures that can disrupt replication and repair, causing instability with qualitatively distinct properties[8,9]. Furthermore, even amongst repeats of the same motif, locus-specific properties can introduce variability in the length-dependent rates of expansion and contraction and directional differences in bias (e.g., for long CAG repeat loci[7]). One well-studied example is the CAG repeat locus responsible for Huntington's disease. A recent analysis showed that a secondary phase of expansion-biased, accelerated instability rates best explains somatic repeat expansion and its association with disease progression[56]. This locus-specific inference does not conflict with our observation that contraction bias terminates the bulk of the genome-wide distribution; indeed, this may indicate that, at lengths well-above those studied in the present manuscript, additional directional flips in bias may occur. This, along with potential inter-locus variability, may contribute to the modest number of repeats at lengths above the truncation point.

Our analysis offers a potential explanation for the prevalence of repeats at lengths that risk progression to disease. First, the dynamics of short and long repeats decouple due to the rapid onset, and subsequent dominance, of repeat length instability. Short repeat dynamics are dictated by substitutions alone, such that repeats within this regime are roughly indistinguishable from random strings of nucleotides of the same length. Longer repeats are primarily subject to distinct mutational forces, exhibiting rapid expansions and contractions and a higher rate of repeat fission, which increases the total number of repeat tracts. Amongst long repeats, those of mid-length primarily experience substitution-based fission, while mutations in the longest repeats are effectively substitution-independent (i.e., fission is driven by non-motif insertion). This is inconsistent with previous literature that suggested substitutions prevent disease by providing a stopping force that counteracts indefinite expansion[44,48,54,57–59]; instead, our analyses suggest this is primarily a consequence of contraction bias at long tract lengths (similar to previous proposals based on very early data[24,26,32,46]). Given the negligible role of substitution, there is little overlap in the mutational forces—and, subsequently, the underlying mechanisms—between the shortest and longest repetitive sequences included in our analyses. In this sense, long repeats emerge as independently evolving genomic elements (with parallels to the concept of selfish genetic elements[60–62]). Monotonically increasing instability rates generate length-dependent dynamics under which expansions lead to further instability, while decreasing length is effectively stabilizing; the former results in frequent forays into long length bins that may be the precursors to disease. The onset of this process leads to a natural definition for the shortest 'unstable' repeat (roughly 6−12 nt, far below disease length). This dynamical definition is distinct from measuring the lowest length where expansion or contraction rates start exceeding the background indel rate (as low as two units for many motifs; Fig. 3b and Supplementary Fig. 5), which may better inform the molecular underpinnings of repeat instability. This difference in scientific goals underlies the debate in the literature concerning the definition of unstable repeats[63].

Provided selection plays little role in directly modifying repeat length, the conservation of the distribution in steady state implies that the underlying mutational mechanisms (i.e., DNA replication and repair) are highly conserved. Generically, such mechanisms play a broad role in maintaining sequence fidelity of the entire genome, primarily preventing single-nucleotide mutations; due to the substantially larger target size, it is unlikely that machinery responsible for both single-site mutations and instability-driven length changes are optimized to properties of the latter. The abundance of long repeats may thus be an inescapable consequence of the pleiotropic function of the machinery maintaining genome-wide sequence fidelity.

It remains unclear which biological mechanisms control the key properties of repeat length instability described in our study. The proposed mechanism(s) should be able to explain length dependencies of instability rates (Fig. 3b and Supplementary Fig. 5) that show: (a) rapid onset from ~6−12 nt, surpassing the rate of substitutions, (b) greater-than-linear increase in the expansion/contraction rate per target above ~10 nt, (c) generically asymmetric rates of expansion and contraction with initial expansion-bias, followed by terminal contraction-bias, (d) single-unit expansions/contractions, regardless of tract length, and (e) parallel expansion and non-motif insertion rates, suggesting a common origin (Fig. 3a and Supplementary Fig. 4). Surprisingly, these observations appear to be largely independent of both

motif sequence and unit length (Supplementary Fig. 5), suggesting a common biological origin.

Two widely studied mechanisms, replication slippage and mismatch repair (MMR), likely explain part of the story[8,9,64–68]. Slippage, when newly synthesized DNA partially unwinds and realigns out of register, should strongly depend on the unit length; however, we see only minor variation associated with unit length (Supplementary Fig. 5). While slippage during DNA replication produces loop-outs on both strands symmetrically[67], subsequent small loop-processing by MMR preferentially results in contractions[69] due to bias towards the nascent strand[70]. Slipped-strand structures may be a motif-independent source of loop-outs subject to the same MMR-processing; in contrast, other secondary structures are motif-specific and therefore cannot be the primary source of repeat instability but can potentially explain differences between motifs[71] (Supplementary Fig. 1b). Importantly, the observation of mostly single-unit expansions/contractions argues against mechanisms involving larger structures (e.g., long hairpins that cannot be processed by MutS$\beta$[72–77]), as these would be expected to generate multi-unit indels.

Single-unit expansions have also been observed in a different context: Okazaki fragment maturation by flap-endonuclease *FEN1*[78]. Imprecise removal of the flap formed by the displaced 5′-flank of an Okazaki fragment may lead to expansion bias[79] and introduce an associated length scale. A secondary mechanism takes over when flaps exceed 30 nt[80]; speculatively, long repeats could give rise to long flaps. Likewise, another flap-endonuclease, *FAN1*, which recently emerged as a genetic modifier of several repeat expansion disorders, was implicated in the processing of various slip-outs and demonstrated differential activity depending on flap length[81]. Altogether, this illustrates how different mechanistic explanations may apply to repeats of distinct lengths, generating emergent properties like length-dependent expansion-contraction bias.

In addition to advancing a mechanistic understanding, substantial effort continues to be dedicated to both assembling datasets and developing estimation techniques specific to repeat instability, due to the inherent difficulties associated with repetitive DNA. Given the difficulty of this task, the present work demonstrates how direct rate estimates can be informed by orthogonal data. The comparative robustness of estimates of the distribution of repeat lengths provides constraints on properties of instability that can serve as a new means for evaluating the quality of differing rate estimates. The DRL may also serve as a summary statistic informative about the evolutionary history of mutation rates and mechanisms, including in species where no population data exists. Indeed, our rapidly improving understanding of repetitive elements, which have historically evaded sequencing efforts, unlocks a range of new questions about the composition and evolution of the genome.

## Methods

### Genome sources
Genome fasta files for T2T-CHM13_2.0 were downloaded from UCSC: http://hgdownload.soe.ucsc.edu/downloads.html#human. Alternate human assemblies and mammalian genomes were downloaded from the NCBI genome database: https://www.ncbi.nlm.nih.gov/datasets/genome/.

### Motif labeling
Throughout the present study, repeat motifs are given a standardized label according to alphabetical order within the list of all cyclical permutations of a given motif (e.g., CAG, AGC, and GCA) and their reverse complements (e.g., CTG, GCT, and TGC). Outside of coding regions, cyclical permutations of a motif become mostly indistinguishable, both bioinformatically and biologically (after exceeding some minimal length relevant to processes such as protein binding site recognition). Likewise, if not considering specific hypotheses such as transcription

direction, reverse complementary motifs should be treated as equivalent because Watson and Crick strands are indistinguishable outside of telomeres. The present study does not investigate any of these specific biological hypotheses, and so we combine results for all equivalent motifs under a single label to increase statistical power. In this arrangement, well-studied motifs may receive a label that differs from that commonly used in the literature (e.g., Huntington's disease (CAG)$_n$ repeats and myotonic dystrophy (CTG)$_n$ repeats are both labeled 'AGC').

### Generation of empirical repeat tract length distributions
Repeat tract length distributions were generated by counting consecutive complete motifs (i.e., perfect motifs, no interruptions, and no partial motifs). Each distribution was assembled by counting contiguous, uninterrupted repetitions of a specified motif. Instead of introducing an arbitrary tolerance for interruptions when counting repeats of a given length, this strict definition allows for straightforward bioinformatic assembly of the DRL. The regex pattern '([ATGC]{1,6}?)\1+' detects arbitrarily long tracts of repeated nucleotides, finding any motif with a unit length of 1–6 nt. Using a 'regex' implementation in Python 3 (pypi.org/project/regex/, version 2024.11.6), all motifs can be detected simultaneously by using the 'finditer' command with the 'overlapped = True' option. Because this pattern detects repetitions of motifs, separate regex patterns were used to detect single instances of each motif (i.e., $L = 1$), taking the form '([ATGC]{n})\1{0}(?!\1)', where $n$ is each motif unit length 1–6. The results of all regex searches were combined to generate a histogram of counts for each motif (pooled under the appropriate label) at all tract lengths present in the genome (i.e., the DRL). Histograms representing counts of non-motifs (i.e., the lengths of contiguous regions where a particular motif is absent; required for computational modeling) were generated on a per-motif basis, using the regex pattern '(?:(?!' + motif + ')[ATGC])+' and combining the results for all cyclical and reverse-complementary permutations of the given motif.

Bootstrap confidence intervals were generated around the T2T-CHM13 repeat length distribution. The genome was divided into 1 Mb contiguous non-overlapping segments, discarding any sub−1Mb chromosome ends. DRLs were measured for each segment. A distribution for the full-length genome was then reconstituted by randomly sampling from these segments, allowing replacements, and summing the distributions from each segment. This process was repeated 1000 times and 95% confidence intervals were generated by separately taking the minimum and maximum in each length bin after removing the top and bottom 25 counts.

For the various mammalian genomes, the same counting procedure was applied. Assemblies generated from short-read sequencing frequently contain many short contigs, which typically originate from poorly sequenced regions containing transposable elements; any contig of length < 10 kb was discarded. Taxonomic data were retrieved from https://ftp.ncbi.nlm.nih.gov/pub/taxonomy/. The median distribution of a given taxonomic group was assembled by gathering the normalized DRLs (see below) for every member of the group (i.e., for primates, this includes humans, and for mammals, this includes primates) and taking the value of the median species for each length bin.

### Distribution normalization
After initially computing the DRL for each motif from the T2T genome, we sought to compare the shape of each histogram of raw counts to those assembled from distinct human reference assemblies (Supplementary Fig. 1c) and from references for various species (Fig. 1b and Supplementary Fig. 2). To compare distributions estimated from assemblies with differing total target size, it was necessary to normalize each distribution (i.e., divide by the total number of counts, summing over length bins) to standardize the overall scale. We refer to the normalized DRL as the probability distribution of repeat lengths,

which we interpret as an estimate of the probability of randomly sampling a repeat of length $L$ from the set of all contiguous motifs (including $L = 1$) in the assembly; when specific length classes are omitted, this becomes a related conditional probability distribution (i.e., $P(L|L > L_{min})$). Shorter assemblies (particularly those with lower quality and read depth at repetitive loci) have a reduced overall number of sequenced repeats and a threshold for statistical (and potentially stochastic dynamical) noise at a lower length. To ensure we are comparing estimates robust to statistical noise, we truncate each DRL above the lowest length bin containing less than 30 counts. This results in otherwise comparable normalized DRLs (assuming the same motif) with distinct truncation points based on non-normalized counts. Qualitative differences between the shape of the resulting normalized DRLs in remaining comparable length bins are indicative of differences in the evolutionary parameters (e.g., mutation rate, selection, etc.) or systematic error profiles (or both) between compared assemblies.

In addition to normalization for empirical comparisons, the empirical DRL and parameterized theoretical DRLs generated by our computational model were normalized by summing only over length classes above a specified minimum length to produce a comparable normalized DRL, conditional on $L \geq L_{min}$. This improved the summary statistic used to characterize differences between these distributions. Further details and justification for the specific choice of $L_{min}$ are provided in the *Bayesian inference procedure*, below. To make figures easier to interpret, normalized DRLs from the computational model were subsequently rescaled (where noted) to match the non-normalized counts for T2T-CHM13 by multiplying each normalized DRL by the sum of counts for bins $L \geq L_{min}$ in the T2T-CHM13 DRL.

## Bioinformatic estimation of substitution and indel rates

De novo mutation datasets were acquired as VCF files (or equivalent) from various published sources[82–89], representing a total of 10,912 parent-child trios with available SNV data and 9,387 trios with available indel data. This dataset was compiled in McGinty and Sunyaev, 2023[90], and comprised of all freely available trio samples at the time of analysis; samples from distinct VCFs were pooled to increase statistical power. We assumed that all individuals have the same underlying mutation rates. Variants were mapped to GRCh38 either in the original study, or subsequently, using 'pyliftover' (pypi.org/project/pyliftover/, version 1.3.2). The average substitution rate was estimated to be $1.2 \times 10^{-8}$, calculated as: number of substitutions/approximate number of sequenceable nucleotides in the diploid genome (see below)/number of offspring genomes in the dataset. We classified substitutions according to six categories based on trinucleotide context and the motif in question, as follows: for the example of mono-A motifs, using B to represent non-A nucleotides, we determined rates (in parentheses) of ABB > AAB and BBA > BAA ($4.58 \times 10^{-9}$) representing repeat-lengthening events, AAB > ABB and BAA > BBA ($7.74 \times 10^{-9}$) representing repeat-shortening events, ABA > AAA ($2.74 \times 10^{-9}$) representing fusion events, AAA > ABA ($4.35 \times 10^{-9}$) representing fissions, BBB > BAB ($3.80 \times 10^{-9}$) representing the rate of $A_1$ creation, and BAB > BBB ($6.17 \times 10^{-9}$) representing the loss of $A_1$. Rates for all other B-substitution processes were not estimated.

We calculated indel rates as a function of repeat tract length. Using positional information, upstream and downstream sequences for each event were pulled from the reference genome, under the assumption that the sequence of the parental genome is identical to the reference genome. For every focal motif, we used the reference sequence to determine tract length. Indel rates per tract length per motif were estimated by dividing by the number of repeats of that length, obtained by generating DRLs in a GRCh38 genome masked for low-quality regions (see below). Each indel was classified as an expansion, contraction, or non-motif insertion, additionally measuring how many motif units were added/removed in the event. We limited

mutations in our computational model to +1/−1 unit changes in length at appropriate rates. We also measured the rate of indels for all B positions (with respect to each motif; mono-A rates in parentheses), separately estimating the rates of BB > BBB ($1.38 \times 10^{-10}$), BBB > BB ($4.37 \times 10^{-12}$), and ABA > AA ($2.76 \times 10^{-10}$) events. Because B strings were not modeled as having length-dependent instability, we measured the average rate, i.e., the rate per unit.

Limitations of the VCF file format, namely the lack of any information at unmutated positions, force the treatment of the pooled-trio VCFs as a complete record of variants in all individuals. At a coarse level, this problem was minimized by assuming that 100 kb regions lacking any substitutions across the combined dataset suffer from regional mappability issues. These regions were masked in GRCh38 when estimating the denominator for rate calculations. At the fine level, this issue persists: mutations may have been filtered (prior to populating the VCF file) due to localized drops in sequencing quality, resulting in false negative calls and undercounting in the pooled estimates. This results in an underestimate of mutation rates, because counts from GRCh38 used in the denominator remain static. This may particularly affect estimation of instability rates as repeat tract length increases, because long repeats are known to interfere with several facets of the sequencing and bioinformatic processes[34–37]. We believe this systematic error mode, leading to progressively more severe underestimation of instability rates with increasing tract length, is the underlying cause of non-monotonicity observed in these rate estimates (Supplementary Fig. 5). Mononucleotide repeats may be especially susceptible to systematic rate underestimation, as they are among the most difficult motifs to sequence[37].

The popSTR repeat instability dataset, representing 6,084 parent-child trios, was acquired from the supplement of Kristmundsdottir et al., 2023[45]. This dataset was incorporated into our inference due to the unique methodology, which provided high-quality calls of mutations extending beyond short tract length repeats that allowed us to produce length-stratified rate estimates. Files 'bpinvolved_extended' and 'mutRateDataAll.gz' were downloaded from https://github.com/DecodeGenetics/mDNM_analysisAndData. Due to our focus on uninterrupted repeats, we measured the longest contiguous repeat tract within the provided coordinates for each event. We limited the dataset to loci where the popSTR-reported reference tract length agreed with our own measurement in GRCh38. The 'bpinvolved_extended' file contains a mix of phased and unphased data; where the parental length for a given mutation was not assigned by phasing, we assumed that it originated from the parental copy, which minimizes the difference in tract length between the proband repeat and any of the parental repeats. Skipping this phasing step under the assumption that all events originated from the reference length allele (but retaining the size and direction of the event), as we do for the pooled-trio dataset, results in relatively minor differences in counts per length bin. The 'mutRateDataAll.gz' file contains information on the number of trios where all three samples passed sequencing quality filters at a given locus, and the length of the repeat tract at each locus in GRCh38, but lacks information on the parental genotypes for each of these loci (i.e., the file does not report pass/fail counts stratified by parental tract length). For the denominator of the popSTR mutation rates, we thus generated a distribution of passing counts (using the reference length for each locus), multiplied by two parental alleles. This assumption leads to some amount of misestimation of rates: loci containing long repeats show higher tract length variance in the population (due to higher instability rates), and thus individuals are more likely to differ from the reference genome. It is unclear whether a related effect (owing to the absence of loci with reference tract lengths below 10 nt in the popSTR dataset), or some other unknown error mode, is responsible for the apparent overestimation of popSTR-based rates at shorter tract lengths where direct comparisons to reliable pooled-trio estimates are possible (Supplementary Fig. 5).

We note that the popSTR dataset differs from the pooled trio data in several aspects: the popSTR caller provides no estimates for reference tract lengths below 10 nt; mutations are classified by length change, failing to distinguish between expansions and non-motif insertions; and, due to data access limitations, we were unable to assess the nature and magnitude of potential systematic errors detailed above. These distinctions precluded direct merging of instability rate estimates with the pooled trio data; popSTR-based estimates were instead incorporated into our inference by informing the prior (see below). Insertion rates, which are the sum of expansion and non-motif insertion rates, were used as a surrogate for expansion rates under the assumption that non-motif insertion remains far more infrequent than expansion (consistent with estimates from the pooled trio data at roughly 1% of total insertions).

For substitution and indel rate estimates based on either the pooled-trio or popSTR datasets, we calculated 95% confidence intervals based on 200 Poisson samples of the mutation counts, removing the top 5 and bottom 5 values per length bin (see Fig. 3b and Supplementary Fig. 5). We note that error bars provided on each estimate represent only Poisson distributed statistical error bars associated with point estimated counts in the numerator of each rate and are therefore subject to the above and any additional systematic errors underlying variant calling.

## Computational modeling of repeat length dynamics

We used a custom-written script in Python 3 that models repeat dynamics by directly manipulating the distribution of repeat lengths. We simultaneously tracked and manipulated the length distribution of B strings. As detailed above, we assumed a binary genome consisting only of A and B sites, where A is a repeat unit and B represents any non-A unit; as a result, B strings do not a priori represent repetitive sequences. Mutations are applied in aggregate such that, in each generation, repeats transition between integer length bins according to rules associated with each mutational process, while the B distribution is updated accordingly (e.g., a substitution that lengthens a repeat simultaneously shortens a B string). Mutation rates were restricted to be sufficiently low to model only a single mutation event per repeat per generation. The non-normalized distribution was evolved and subsequently normalized to create a probability distribution for comparison to empirical data. This approach is far more computationally efficient than simulating an entire genomic sequence, subsequently applying mutations and generating a distribution; computational time in our script scales with the number of length bins rather than with the length of the genome. Tracking only the distribution discards information about the location of particular mutations, instead generating an expected number of mutations for each category per length bin per generation. Except where specified, we used a deterministic approximation to assess the behavior of the expectation value of each bin as the distribution evolves toward steady state via repeated application of the mutation kernel. To understand the impact of stochastic fluctuations on the steady state distribution, we additionally implemented a model that represents fluctuations by Poisson sampling the expected change to each length bin per generation. We model stochastic fluctuations around the applicable rates by sampling mutational counts, but without constraining individual transitions (i.e., a net number of mutations may leave a given class, but the number introduced elsewhere, as a result, is appropriately distributed only on average due to an independent sampling procedure). All subsequent analyses were performed using the deterministic results, as modeling independent fluctuations in each bin showed no qualitative differences (Supplementary Fig. 13e).

Mutations affect the distribution via the following well-defined rules for substitutions and indels (see Fig. 2 for illustration). These rules assume that each mutation adds, subtracts, or substitutes a single, complete repeat unit (i.e., the most prevalent class of length changes, seen in Fig. 3a and Supplementary Fig. 4). Using the example of a repeat of $L = 6$, a lengthening substitution subtracts one count from the $L = 6$ bin and adds one to the $L = 7$ bin. A shortening substitution subtracts one from the $L = 6$ bin and adds one to the $L = 5$ bin. A substitution causing repeat fission subtracts one from the $L = 6$ bin and adds two new repeats, either one $L = 1$ and one $L = 4$, or one $L = 2$ and one $L = 3$ (when evolving the distribution in aggregate, both occur simultaneously with appropriate relative rates). The reverse process of fission is fusion, in which an $L = 6$ repeat can be generated by fusing an $L = 1$ with an $L = 4$, or by fusing one $L = 2$ and one $L = 3$ repeat, while the mutated B unit is replaced with an A unit and added to the repeat length. Lengthening and shortening substitutions act locally (i.e., counts leave the $L$ bin and move to the adjacent $L + 1$ and $L - 1$ bins, respectively). Substitution of an $L = 1$ in the A distribution also corresponds to fusion of B strings; the reverse, i.e., substitution of a length one B string, generates fusion in the A distribution. Fission and fusion substitutions inherently act non-locally in length space: fission results in the loss of one count in the $L$ bin and gain of two counts that are evenly distributed across all bins of length $\leq L - 2$; fusion evenly subtracts two counts from bins $\leq L - 2$ to add a count to $L$. The net effect of substitutions conserves the total length of the genome, i.e., the sum of the length of all A repeats plus the sum of the length of B strings remains constant under substitutions alone.

The rates of lengthening, shortening, fission, and fusion substitutions per generation are separately estimated using the three-unit context: BBA > BAA (or ABB > AAB) for lengthening substitutions, AAB > ABB (or BAA > BBA) for shortening substitutions, AAA > ABA for fissions, and ABA > AAA for fusions. All substitution rates were assumed to be independent of repeat length, based on our previous observations showing little to no rate increase with increasing repeat length[90]. The target size for lengthening substitutions is two per repeat (i.e., the two sites adjacent to each repeat boundary). Likewise, the target size for shortening substitutions is also two per repeat, representing the two boundary units of the repeat (assuming $L > 1$). The target size for fission substitutions is $L - 2$ per repeat, representing all non-boundary units within the repeat. The target size for all fusion events is proportional to the $L = 1$ count of the B distribution. Equations governing these processes are described in detail in the Supplementary Note.

Indel mutations operate under an analogous logic, but with a few important distinctions. Indels, by definition, do not conserve the length of the genome. Expansions and contractions act strictly locally, but the location of the event is indistinguishable within the repeat, affecting any of the units rather than just the boundaries; this results in a per-repeat target size $L$ for these mutations, rather than 2. Non-motif insertions (i.e., AA > ABA) cause fission, resulting in the loss of one count in the $L$ bin and gain of two counts that are evenly distributed across all bins of length $\leq L - 1$; deletion of a B string of $L = 1$ (i.e., ABA > AA) causes fusion, which evenly subtracts two counts from bins of length $\leq L - 1$ and adds one count to bin $L$. Indel rates for expansions and contractions are incorporated in a length-dependent manner, described above, in contrast to substitution rates. We did not model length dependence for B indels, as most B strings represent a combination of nucleotides and not necessarily STRs with any biological relevance. This assumption should not impact the evolution of the A distribution after normalization, which is only coupled to the $L = 1$ class of the B distribution; this length class is dominated by substitution rate dynamics and not subject to repeat instability.

**Time-rescaling using a constant speed-up factor.** Due to the large number of iterations required to reach a steady-state DRL, propagating the mutational process directly was computationally prohibitive. Instead, we approximated the DRL by first rescaling time by multiplying all mutation rates by the same constant $10^r$ such that each

iteration represents $10^r$ generations of evolution for a total of $T = 10^{Ir}$ generations (assuming a constant $r$ at all time points run for $I$ iterations); $r$ was limited to integer values for convenience. This defines a set of time-rescaled substitution, expansion, contraction, and non-motif insertion rates as a function of length.

Due to the rapid growth of instability rates with increasing length, these rates quickly saturate, reaching probabilities of one at some length $L$. To avoid multiple mutations per repeat per iteration, we defined a saturation length $L_{max}$ as the length at which the sum of all mutation rates first exceeds 0.1 (e.g., $L_{max} \to \infty$ if the sum of rates remains below 0.1 at all lengths). $L_{max}$ thus demarcates the linear mutation regime, below which multiple mutation events remain rare. In addition to dependence on instability rate parameters, $L_{max}$ is dependent on $r$: increasing the speed-up factor increases mutation rates by $10^r$, which decreases $L_{max}$. To ensure reasonable computational time, $L_{max}$ was limited to a maximum of 200 (i.e., $\min\{L_{max}, 200\}$, computed separately for $A$-repeats and $B$-string lengths), which extends well beyond the empirical DRLs. To prevent loss of mass associated with the finite length grid, we imposed a reflective boundary condition at $L_{bound} = \min\{L_{max}, 200\}$ (i.e., all transitions from lengths $L < L_{bound}$ to $L \geq L_{bound}$ were assigned to $L_{bound}$). This results in artefactual behavior near the boundary but provides a reasonable approximation when the expected number of counts drops below one at lengths far below $L_{bound}$. Substantial counts at the boundary are indicative of unrealistic distributions (often associated with diverging total genome size), provided $L_{bound}$ is sufficiently far from the maximum well-populated lengths in the comparable empirical distribution.

For a given parameterization, producing a grid of DRLs requires choosing a constant speed up $r$ and the boundary length $L_{bound}$ appropriate for each parameter combination (the required computational time is largely determined by the number of parameter combinations with the lowest value of $r$). This procedure can be used to produce a coarse grid of parameters (e.g., for comparison to alternative approximations), but it proved computationally prohibitive for the dense grid needed for inference.

**Step-wise speed-up procedure.** To produce finer grids of DRLs (for several parameterizations), we implemented a procedure that reduces overall computational time, while producing approximately the same DRL as that under a constant speedup (described above). This procedure models the evolution of the DRL by performing several, discrete phases of evolution, each with successively smaller time-rescaling factors $10^r$. Each stage is allowed $10^6$ iterations of evolution under the specified $r$ (and the associated reflective boundary at $L_{bound} = \min\{L_{max}, 200\}$ for each parameter combination); $r = 3$ for the first stage, and is reduced to $r = 2, 1,$ and $0$ for subsequent stages. $L_{bound}$ is altered at each stage to maintain the linear mutation regime (i.e., ensuring $L_{max} \geq L_{bound}$). In total, parameter combinations can experience up to four stages (equivalent to $4 \times 10^6$ iterations, or $1.111 \times 10^9$ generations).

For computational efficiency, we first separated parameter combinations that rapidly equilibrate in a single stage of $10^6$ iterations under a sufficiently large rescaling factor $r \geq 3$ (easily identified by $L_{max} \geq L_{bound} = 200$; equivalent to propagation using a single, constant speed-up factor). These were each run at the largest allowed integer $r$ for the equivalent of $10^9$ generations and removed from the grid. All other parameter combinations were subjected to several stages with progressively decreased $r$; after each stage, parameter combinations deemed equilibrated were removed from the grid (again identified by $L_{max} \geq 200$, determined by the preceding $r$-rescaled instability rates). For parameter combinations with $L_{max} < 200$ in the absence of any speed-up factor ($r = 0$), counts in all length bins between $L_{max}$ and 200 were set to zero prior to analysis of the DRL.

To ensure that the multi-step procedure provides a reasonable approximation to the DRL produced under a constant speed-up, we compared inference results over a coarse grid of parameter combinations and found negligible differences (Supplementary Fig. 13). Intuitively, this procedure takes advantage of the faster mutation rates at longer lengths, which equilibrate much more quickly than shorter length bins.

**Initial conditions at $t = 0$.** The computational model was initialized with an initial distribution that is approximately geometric (created by propagating substitutions alone, setting all instability rates to zero) for the equivalent of $10^{10}$ generations (using the largest allowable rescaling, $r = 5$). Using this approximation to the substitution-only steady-state as a pre-simulation substantially reduces equilibration times because the lowest length bins (i.e., those dominated by substitutions) require the most time to equilibrate due to low mutation rates.

Although the eventual steady-state DRL should not depend on the initial state, the choice of initial distributions can dramatically affect equilibration times. We confirmed that the final timepoint DRLs are effectively independent of choice of initial condition by comparing the results of two distinct initial conditions with similar equilibration times (geometric vs geometric plus uniform; Supplementary Fig. 12), finding only minor differences in the deterministic late-time distribution.

**Computational model inputs and outputs.** The script relies on the following as inputs: an initial distribution for A and B (i.e., motif and non-motif) repeat lengths, per-target substitution rates in three-unit context, and per-target mutation rate curves for expansions, contractions, and non-motif insertions. Substitution rates and length-dependent indel rate curves are imported from external files (see above for estimated substitution rates, below for generation of parameterized rate curves); these files, along with the initial repeat length distribution table, can be replaced with appropriate tables for other purposes, if desired. This table must specify rates for each mutational process at all lengths intended to be computationally modeled (i.e., from 1 to $L_{bound}$). For normalized length distributions that reach steady state, the initial distribution can be chosen arbitrarily, in principle, but any specific choice affects equilibration time; due to equilibration time differences, minor differences between the deterministic late time distributions arise from distinct initial conditions (see Supplementary Fig. 13 for comparison between two initial distributions).

Stochastics can be introduced using a command line option to model fluctuations in the mutational process; the number of mutations in and out of each length bin are separately Poisson-sampled (using *numpy.random.poisson*, version 2.2.1) around the expected number of mutational counts in each iteration.

After each run, we output a file containing repeat length counts reported at various time points to show the temporal evolution of the distribution. We subsequently normalized the resulting distributions by dividing each length bin by the total number of repeats in the distribution (see Methods on normalization).

The relative contribution of each mutational force was assessed by producing a single-generation plot of the transitions in and out of each length bin at the final time point (i.e., once steady state was reached, if applicable). To produce these plots (see Fig. 6), we applied the mutation kernel for a single generation and separately computed the number of fissions, fusions, and local changes due to substitutions and indels. For each length bin, the magnitude of total flux in and out was normalized to one. Length bins that have equilibrated should contain equal fluxes in and out; steady state occurs only when all bins show equilibrated fluxes.

**Bayesian inference procedure.** Given our observations indicating a stable distribution of repeat lengths over phylogenetic time scales, we sought to identify mutation rates capable of explaining this

observation. To study the extent to which mutational processes alone can recapitulate the repeat tract length distribution, we constructed a Bayesian inference framework to compare models (i.e., parameterizations) of the length-dependent rates of repeat instability. Each parameterization describes the length-dependent rates of expansion and contraction via a simple functional form; as discussed above, substitution rates are assumed to be length independent in all cases. Within the Bayesian framework, a prior probability distribution on the parameter space is specified (several priors were used for interpretation) and used to weight the likelihood to calculate a posterior probability that a given parameter combination accurately describes the length dependence of the repeat instability rates. In the present setting, the likelihood is constructed by comparing the empirical repeat tract length distribution to the late-time distribution generated by computationally modeling a given parameter combination. Due to the analytic intractability of this likelihood, we used ABC[55] to approximate the posterior probability distribution, which avoids specifying the likelihood explicitly. Additionally, length bins are presumably correlated due to the complex mutational transitions underlying the distribution, complicating naïve construction of the likelihood. In contrast, ABC-based inference circumvents this issue by approximating the posterior in terms of summary statistics that appropriately characterize the DRL (see discussion of summary statistics below). After specifying summary statistics, the late-time distribution for each parameter combination was summarized for comparison to the empirical distribution (e.g., mononucleotide A repeat tract lengths in T2T-CHM13). For each parameterization, we specified a discrete grid of parameters for comparison to the empirical distribution, the result of which was weighted by the prior probabilities for those parameters (equivalent to randomly sampling the prior as prescribed in ABC[55]; see below for specific priors used) to compute the posterior.

We chose to use the KL divergence, a well-established statistic for distribution comparison, to characterize the difference between the empirical ($P_L^{emp}$) and parameterized repeat tract length distributions. The KL divergence quantifies the extent to which each parameterized distribution diverges from the empirical distribution and was calculated for all parameter combinations (denoted $\theta$ below) on the discrete grid using the following definition.

$$D_{KL} = \sum_{L=L_{min}}^{L=L_{max}} P_L^{emp} \log\left(\frac{P_L^{emp}}{P_L(\theta)}\right) \tag{1}$$

We note that comparing the empirical distribution to itself results in a divergence of zero such that the KL values are equivalent to the difference between the modeled and empirical distributions (i.e., $\Delta D_{KL} = D_{KL}\left(P_L(\theta), P_L^{emp}\right) - D_{KL}\left(P_L^{emp}, P_L^{emp}\right) = D_{KL}(P_L(\theta), P_L^{emp})$). To define a cutoff for ABC rejection, we estimated the divergence between the human empirical distribution and the ensemble of primate genomes using the same statistic. Under the assumption that primates evolved towards the same steady state (i.e., the mutational parameters remain constant across the phylogeny), we proceeded under the assumption that differences between the repeat tract length distributions in distinct species are due to a combination of stochastics and bioinformatic errors due to the lower coverage and short read technologies used to assemble primate reference genomes. Due to the difference in assembly lengths, we added a pseudocount of one to all length classes in all species to avoid divergence of the statistic and confirmed that our results were qualitatively independent of the choice of pseudocount between 0.01 and 100. $L_{max}$ was set to the longest modeled length bin, $L = 200$. We set the lower bound to $L_{min} = 4$ to ensure that the ordering of $D_{KL}$ statistics computed for all primates remains roughly consistent: setting $L_{min} \geq 4$ resulted in the smallest values for the human HG38 reference (which was not included in subsequent analyses) and the largest values for the most divergent primates (i.e., those on the loris branch). We used the range of 36

computed primate KL values to very roughly define a rejection threshold by throwing out the largest 2 values; we considered the remaining 34 values (i.e., roughly, the closest 95% of ranked primates; denoted 'p95' in the following) and used this to approximate the variance of $D_{KL}$ (i.e., $\sigma_{D_{KL}}^2 \approx (D_{KL}^{p95}/2)^2$) associated with stochasticity and sequencing errors.

We approximated the posterior probability (up to a normalization constant) following the prescription in Wilkinson[55] wherein ABC is applied with a soft rejection threshold by rejecting values of $D_{KL}(\theta)$ with the following probability based on the primate-estimated variance.

$$Pr(\theta|\text{data})Pr(\text{data}) = Pr(\text{data}|\theta)Pr(\theta) \sim Pr(\theta)e^{-\frac{D_{KL}^2(\theta)}{2\sigma_{D_{KL}}^2}} \tag{2}$$

Here, $Pr(\theta|\text{data})$ is the posterior probability distribution over the grid of parameter combinations $\theta$, $Pr(\theta)$ is the prior, and the Gaussian falloff is the rejection probability (up to normalization) for a given parameter combination. This soft-rejection procedure provides a slightly better approximation for the posterior distribution than rejection with probability one (e.g., reject all parameter combinations with $D_{KL}(\theta) > D_{KL}^{p95}$, roughly corresponding to the primate-estimated 95% confidence interval). Additionally, Wilkinson argues that the Gaussian rejection probability quantifies model misspecification inherent in the procedure, as all parameterizations (i.e., models) employed herein are imperfect approximations of the true instability rate parameters.

**Model comparison using Bayes factors.** To assess the relative explanatory power of each parameterization $M_\theta$ modeling the length dependence of repeat instability, we computed a Bayes factor for each model $BF(M_\theta)$ using the definition below.

$$BF(M_\theta) = \int d\theta\, Pr(\theta|M_\theta)Pr(\text{data}|\theta, M_\theta) \sim \int d\theta\, Pr(\theta)e^{-\frac{D_{KL}^2(\theta)}{2\sigma_{D_{KL}}^2}} \tag{3}$$

Here, the right-hand side is our previously computed approximation to the posterior, integrated over the parameter space for a given model. As we were only interested in the relative Bayes factor between models, proportionality constants can be ignored, including the overall normalization and an assumed uniform prior over model space. The Bayes factor for a model naturally controls for the number of DoF in each parameterization because integration over the weighted posterior is performed in parameter spaces with differing dimensionalities. Once computed, models were compared by interpreting the Bayes factor ratio (BFR) as indicative of the relative statistical support between two parameterizations of interest.

$$BFR(M_1, M_2) = BF(M_1)/BF(M_2) \tag{4}$$

We then used Jeffreys' scale to interpret the strength of statistical support for each model.

**Parameterizations of repeat instability rates.** We tested several parameterizations to assess consistency with the empirical DRL. We focused on mononucleotide A repeats, as both the distributions and rate estimates were supported by the most empirical data. For computational convenience, we defined a sequence of nested parameterizations (see Supplementary Fig. 23) that could be computed simultaneously across the grid of the parameter combinations under the model with the largest number of DoF. To define the most general set of length-dependent instability rate models, we parameterized expansion and contraction rates, $\epsilon(L)$ and $\kappa(L)$, respectively, as

independent power law functions at all lengths $L > 8$.

$$\epsilon(L>8)=c_\epsilon\left(\frac{L}{9}\right)^{\tau_\epsilon}, \quad \kappa(L>8)=c_\kappa\left(\frac{L}{9}\right)^{\tau_\kappa} \quad (5)$$

Rates for $L = 1$–8 were taken directly from empirical rate estimates; the expansion and contraction rates at longer lengths were parameterized in terms of $c_\epsilon$ and $c_\kappa$, which denote their respective values at $L = 9$ (i.e., the first parametrized length bin). Guided by our empirical estimates, we assumed that the rate of non-motif insertion $\iota(L)$ is directly related to the rate of expansion with the same length dependence at 1% of the rate (i.e., $\iota(L) = \epsilon(L)/100$ at all lengths. This results in instability rates characterized by four independent parameters ($c_\epsilon, \tau_\epsilon, c_\kappa, \tau_\kappa$). We then constructed a series of nested lower-dimensional models for comparison. A natural way to reduce the dimensionality of the parameter space is to introduce symmetries corresponding to $c_\epsilon = c_\kappa$ and/or $\tau_\epsilon = \tau_\kappa$. The simplest model assumes fully symmetric expansion and contraction rates (i.e., both $c_\epsilon = c_\kappa \equiv c$ and $\tau_\epsilon = \tau_\kappa \equiv \tau$) with a two-dimensional parameter space ($c, \tau$). We treat this as a null model corresponding to a frequently discussed[8,19,22,27] biological interpretation of repeat slippage. The parameter space can be reduced to three DoF by restricting to either $c_\epsilon = c_\kappa$ or $\tau_\epsilon = \tau_\kappa$, which need not have straightforward biological interpretations. We constructed an additional 3 DoF model parameterized by ($m, \tau_\epsilon, \tau_\kappa$) by treating the expansion and contraction rates at $L = 9$ as increased by a common multiplier $m$ relative to their values at $L = 8$ (i.e., $\epsilon(9) = m\,\epsilon(8)$, $\kappa(9) = m\,\kappa(8)$). For computational expediency, we embedded this model within the four-dimensional grid of parameters by appropriately choosing intervals for $c_\epsilon$ and $c_\kappa$ when defining the grid discretization.

We used two distinct, non-nested parameterizations for subsequent analyses. To test the reliance of our inference on the functional form of the length dependence (i.e., power-law parameterization), we defined an additional parameterization by replacing the length dependence for $L > 8$ with logarithmic growth in the following form:

$$\epsilon(L>8)=c_\epsilon\left(\frac{\log(L-7)}{\log 2}\right)^{\tau_\epsilon}, \quad \kappa(L>8)=c_\kappa\left(\frac{\log(L-7)}{\log 2}\right)^{\tau_\kappa} \quad (6)$$

Here, the dependence on log 2 ensures that $c_\epsilon$ and $c_\kappa$ parameterize the values at $L = 9$ of the expansion and contraction rates, respectively. Under this parameterization, empirical estimates were again used for all lengths $L \le 8$. This functional form retains monotonicity while growing more slowly at longer lengths to model a saturation-like effect.

We analyzed a second version of the power-law parameterization that extends the functional form to all lengths such that the rates are fully independent of empirical estimates. We re-parameterized the functional dependence in terms of the parameters ($\lambda_\epsilon, \tau_\epsilon, \lambda_\kappa, \tau_\kappa$), where $\lambda_\epsilon$ and $\lambda_\kappa$ correspond to the length at which each instability rate exceeds the relevant substitution rate (i.e., $\epsilon(\lambda_\epsilon) = \mu$ and $\kappa(\lambda_\kappa) = \nu$).

$$\epsilon(L)=\mu\left(\frac{L}{\lambda_\epsilon}\right)^{\tau_\epsilon}, \quad \kappa(L)=\nu\left(\frac{L}{\lambda_\kappa}\right)^{\tau_\epsilon} \quad (7)$$

Here, $\mu$ and $\nu$ are point estimates of the average lengthening and shortening substitution rates, respectively, for a given motif; note that this parameterization is defined at all tract lengths (including $L \le 8$). This allowed us to directly infer the length scale of the instability-substitution rate crossover and assess the extent to which our inferences from the above parameterizations rely on direct use and accuracy of empirical rate estimates below $L = 9$. After confirming that expansion-biased parameters do not approach steady state DRLs, the parameter space was further limited to asymptotically contraction-

biased parameter combinations (i.e., with $\tau_\kappa > \tau_\epsilon$) to limit computational time.

The functional form of each of the aforementioned parameterizations is specified in Table 1.

**Construction of prior distributions.** For each parameterization, we constructed an uninformative prior by treating each parameter combination as equally probable with probability equal to $1/n$, where $n$ is the number of computationally modeled points on a discrete grid. We next generated informative priors using approximations derived from our empirical estimates of the expansion and contraction rates. For power-law parameterizations that include empirical estimates at low lengths (Models 1–5 in Table 1), we performed a linear fit to the popSTR rate estimates (at lengths $L = 11$–29) in log-log space to estimate parameters of best-fitting power laws. Curve fitting was performed in Python 3 (*scipy.optimize.curve_fit()*, version 1.15.1) with the sigma option to specify an array of approximately symmetric log error bars (i.e., approximating a rescaled Poisson as log-normal). We note that Poisson regression does not appropriately model statistical noise due to target size rescaling when estimating rates from mutational counts. We fit using rate estimates at all available lengths. However, we artificially inflated the variance at lengths above and below $L = 13$–21 to model potential systematic errors that generate observed non-monotonicity, likely due to miscalling at the shortest and longest lengths accessible to popSTR. The optimization package produced a covariance matrix for the best-fit line expressed in terms of the slope and intercept in log-log space. We used this covariance matrix to approximate lines representing the 95% confidence bounds around the best-fit line. Using these lines, we estimated the value of the best-fit line and standard deviation at $L = 9$. Assuming no correlation between expansion rate and contraction rate parameters, we used these values to approximate a block diagonal covariance matrix for the parameters ($c_\epsilon, \tau_\epsilon, c_\kappa, \tau_\kappa$) in the four-dimensional model. We then inflated the variance by a constant (100-fold for 'restrictive' informative prior; 1000-fold for 'permissive' prior) and used the rescaled covariance matrix to define a multivariate normal distribution centered at the point-estimates for the best-fit parameters. The informative prior for the four-parameter model was constructed by normalizing this multivariate normal over the discrete grid. Analogous priors for nested models were defined by restricting to the appropriate subset of parameter space, maintaining the relative weights specified by the normal distribution, and normalizing by the number of discrete grid points in this subset.

The approximate nature of our ABC-based inference procedure prohibited the construction of strictly uninformative priors (i.e., Jeffreys priors) for fair comparison between models with differing parametric functional forms. We instead treated the uniform prior as naively uninformative; however, despite the similarity of their parameters, we caution that uniform priors are inequivalent for distinct functional forms. To facilitate very rough model comparison, we constructed a restrictive prior for the logarithm-based model by again fitting the functional form to popSTR rates using *scipy.optimize.curve_fit()*. This produced a point estimate of the best-fit parameters that was used to specify the mean of a multivariate normal distribution. To attempt to define a multivariate normal very roughly comparable to the priors for the power-law parameterizations, we defined the normal distribution in terms of the covariance matrix estimated under the four-parameter power-law model. This comparison relies on the fact that both parameterizations use the same parameters to represent nearly identical quantities (i.e., parameters $c_\epsilon, \tau_\epsilon, c_\kappa$, and $\tau_\kappa$ define constants and exponents in the same way). The procedure described above was then used to define restrictive and informative priors (with 100-fold and 1000-fold inflated variances, respectively) for two- and four-dimensional logarithm-based parameterizations from four-dimensional multivariate normal distributions with appropriately shifted means.

To construct priors for pure power-law models, we again used a uniform prior over the discrete grid to define a naively uninformative prior over the parameter space. Informative priors were defined by again using *scipy.optimize.curve_fit* to estimate best-fit parameters and a covariance matrix from empirical instability rate estimates. Unlike the previous models, rates at all lengths (including $L \le 8$) were fully parameterized by Eq. 7 (see also Table 1); data from both the pooled trio and popSTR datasets were used to estimate the mean and covariance matrix by fitting the functional form to expansion and contraction rate estimates for $L = 4$–15. The covariance matrix was again inflated by a factor of 100 and used to define a four-dimensional multivariate normal around the point-estimated mean values of $(\lambda_\epsilon, \tau_\epsilon, \lambda_\kappa, \tau_\kappa)$.

**Calculation of expectation values from posterior probability distribution.** In addition to identifying maximum posterior probability parameter combinations, we used the posterior distribution to weight various quantities to calculate their expectation values. Expectations of an arbitrary parameter-dependent function $f(\theta)$ were computed as: $E[f(\theta)] = \int d\theta\, f(\theta) Pr(\theta|\text{data})$. Here, the ABC-approximated posterior was used for $Pr(\theta|\text{data})$ after normalizing over the discrete computational grid of parameters $\theta$ such that $E[1] = \int d\theta\, Pr(\theta|\text{data}) = 1$. We used this to compute: the length dependence of the posterior-weighted repeat instability rates $E[\epsilon(L;\theta)]$ and $E[\kappa(L;\theta)]$ for comparison; and the posterior-weighted DRL $E[P(L;\theta)]$ (see Fig. 5).

**Analytic modeling of repeat length dynamics**
To better understand the underlying dynamics that generate the genome-wide repeat length distribution, we attempted to analytically model the effect of each mutational type on the number of repeats at a given length $L$ from first principles. We were interested in describing the steady state distributions that emerge for a subset of parameter combinations, as seen in the results of our computational model. Our goal was to capture the balance between relevant mutative forces, which can vary by repeat length, by writing an appropriate approximation to the steady state equation; the solutions to these equations describe the shape of the normalized repeat length distribution, $P(L)$, restricted to the regime of validity of each approximation. Within this section, we have used the notation $P_L$ to represent the distribution $P(L)$ more compactly when detailing the relevant equations. Each parameter combination defines a functional form for the per-target (i.e., per unit) expansion, contraction, and non-motif insertion rates at lengths $L \ge 9$: $\epsilon(L \gg 1) = \epsilon_0 L^{\tau_\epsilon}$, $\kappa(L \gg 1) = \kappa_0 L^{\tau_\kappa}$, and $\iota(L \gg 1) = \iota_0 L^{\tau_\iota}$, respectively, where the constants $\epsilon_0$, $\kappa_0$, and $\iota_0$ are set by the empirical value of these rates at $L = 8$ and the multiplier $m$ (noting that we set $\tau_\iota \equiv \tau_\epsilon$ to limit the number of free parameters; see inference Methods). Again, these length-dependent rates, in either discrete or continuous form, are denoted with a subscript $L$ (e.g., $\epsilon_L \equiv \epsilon(L)$) in this section for brevity. For substitutions, we refer herein to rates $\mu \equiv \mu_{A \to B}$ and $\nu \equiv \mu_{B \to A}$ for lengthening and shortening mutations, respectively, but later specify separate mutation rates based on three-unit context (e.g., $\mu_{ABB \to AAB}$) when comparing directly to computational model results. While the mutation rates may be well defined by these rates, the combined effect of substitutions and indels on the repeat length distribution requires a description of a number of complicated behaviors, including both local and non-local transitions between lengths across the distribution, non-conservation of the number of repeats due to fission and fusion, and non-linear dependence on the state of the distribution due to fusion (i.e., the generic dynamics are non-Markovian). As a result, our aim was not to describe an exact solution, but instead an expression for the effective dynamics that dominate the maintenance of the distribution in steady state, specifically in the asymptotic regimes associated with the shortest and longest length repeats. Note that this analytic description was motivated by and is strictly applicable to mononucleotide repeat dynamics, where the

species of repeat length-changing mutations are fewer, but the conceptual findings may be generalizable to longer motif repeats (Supplementary Fig. 3).

**Short repeat regime.** First, we focused on the regime of asymptotically short repeats, as their behavior is more straightforward. By assessing the relative rates of substitution and indel processes in the estimated per-target rates (Fig. 2b), one can immediately see that substitutions must dominate the dynamics for the lowest length repeats. Short repeats can be characterized by a straightforward balance between opposing types of substitutions, $\mu$ and $\nu$, which is equivalent to sequence evolution under a two-way point mutation process. At steady state, the resulting distribution is equivalent to the probability of randomly assembling specific strings of length $L$ when the whole genome is randomly sampled between A and B bases with probability $p_A = \mu/(\mu + \nu)$ and $p_B = \nu/(\mu + \nu)$, respectively. The frequency of a length $L$ string of A's (i.e., an A repeat) is geometrically distributed in proportion to $p_A^L$ (i.e., sampling an A, $L$ successive times).

$$P_{L \ll 10} \propto \left(\frac{\mu}{\mu + \nu}\right)^L \tag{8}$$

Here, we have omitted a normalization constant that determines the relative weight of this geometric distribution to the weight of the long repeat tail. For comparison to the computational model (or the empirical distribution), we fixed the normalization constant using the mass of the $L = 1$ bin. The approximation that the effects of expansion, contraction, and non-motif insertion are negligible breaks down at a length determined by the estimated relative rates in Fig. 3b; the regime of validity for this approximation extends roughly to lengths of order $L = 10$.

**Long repeat regime.** The dynamics of long repeats, i.e., for asymptotically large repeat lengths $L \gg 1$, the analysis is complicated by the numerous length-dependent (and parameter-dependent) forces that can potentially contribute to stabilizing the distribution. While expansion and contraction describe inherently local transitions from $L$ to $L + 1$ and from $L$ to $L - 1$, respectively, the effects of non-motif insertions and substitutions on extended repeats are not strictly local. To model this regime, we first wrote a finite difference equation that describes the change in the distribution in a single time step $\Delta t$: $\Delta P_L \equiv P_L(t + \Delta t) - P_L(t)$, where $P_L(t) = P(L, t; \mu, \nu, \epsilon_L, \kappa_L, \iota_L)$ is implicitly dependent on the length scaling of each rate (see Supplementary Note). From this discrete equation, we derived a partial differential equation (PDE) in the large-length continuum limit $\Delta L = 1 \ll L$ that approximates the dynamics in the large length regime (derivation provided in the Supplementary Note). This PDE includes explicit terms depicting the combined local effects of repeat instability due to expansion and contraction, each occurring at distinct length-dependent rates, and the separate effects of repeat fission and fusion, each introducing an integral that captures the aggregate effects of non-local transitions in length. Expansion and contraction collectively generate both symmetric (i.e., bidirectional) and asymmetric local length transitions, which correspond to a diffusion term represented by a second derivative and directional flux term expressed as a first derivative, respectively, each appropriately accounting for length-dependent rates.

While local effects from substitutions and non-motif insertions exist (specifically, transitions $L \to L + 1$ or $L \to L - 1$), as well, they are negligible in comparison to expansion and contraction due to their low relative rates at long lengths and finite target size of two per repeat. Fissions due to both substitutions and non-motif insertions were accounted for as separate non-local contributions to the change in $P_L$. Importantly, the probability of

fission due to substitution is proportional to the target size $(L - 2) \approx L$; for insertions, the rate itself harbors an additional length dependence such that the per-repeat rate of fission scales as $L^{1+\tau_\epsilon}$. As a result, the relative importance of fission compared to local contributions is highly dependent on the parameters $\tau_\epsilon$ and $\tau_\kappa$; similarly, the relative importance of substitution- and insertion-based fission are parameter dependent due to distinct dependencies on length. Thus, a unified description across parameter space requires the inclusion of fission in full form and captures all four mutational effects. While we were able to explicitly describe the integral effects of length changes due to repeat fusion in the continuum (see Supplementary Note), the inherent non-locality is additionally complicated by the non-linearity introduced by pairing two repeats randomly sampled from the distribution. To make further progress, we proceeded under the assumption that fusion remains subdominant at large lengths, which we confirmed via our computational model to be generically true across parameter space. Stochastic fluctuations in the mutation rates were omitted, resulting in a deterministic approximation for the expected repeat length distribution.

Next, we imposed the assumption of steady state (i.e., $dP/dt = 0$), reducing the PDE to an ordinary differential equation (ODE) in length to solve for the shape of the distribution in equilibrium. Despite excluding complications from fusion, the remaining approximation to the steady state equation is, strictly speaking, a second-order integro-differential equation, for which no explicit closed-form solutions exist. The following equation approximates the steady state dynamics in the absence of fusion (i.e., when fusion is subdominant). Here, $\partial_x$ represents a derivative with respect to $x$ (noting that partial derivatives with respect to $L$ become total derivatives in steady state), and $P_L$ is the steady state value of the continuous repeat length distribution at large length $L \gg 1$ up to an overall normalization constant (along with an arbitrary constant set to zero). Again, all continuous functions describing mutation rates (e.g., $\epsilon_L, \kappa_L$) are expressed here as per-target rates.

$$\frac{dP_L}{dt} = 0 \approx \frac{1}{2}\partial_L^2\left[(\epsilon_L + \kappa_L) L P_L\right] - \partial_L\left[(\epsilon_L - \kappa_L) L P_L\right] - (\nu + \iota_L) L P_L + 2\int_L^\infty d\lambda \, (\nu + \iota_\lambda) P_\lambda \tag{9}$$

In order from left to right, the terms appearing on the right hand side describes: length-dependent diffusion (arising from local transitions due to expansion and contraction), a length-dependent local directional flux (due to the bias between expansion and contraction), a net loss of due to fissions that break up length $L$ repeats (i.e., substitutions or insertions that interrupt the repeat sequence; referred to herein as *fission out*), and a net gain due fissions of repeats longer than $L$ (referred to as *fission in*). Fission in represents the sole integral effect, which substantially complicates our analysis; elimination of the integral dependence is discussed below and results in a third-order ODE that maps to this second-order integro-differential equation.

**Contraction-biased rates stabilize the distribution.** Importantly, we found that steady state could only be reached for the subset of parameter combinations with $\tau_\kappa > \tau_\epsilon$, corresponding to cases for which local transitions are asymptotically contraction-biased: $\lim_{L\to\infty}(\kappa_L - \epsilon_L) > 0$ (note that the edge case where $\tau_\kappa = \tau_\epsilon$ is asymptotically expansion-biased based on observations at $L = 8$ and implications of our parameterization). We therefore denote this as the contraction-biased regime, which is characterized by defining the variable $\Delta\tau \equiv \tau_\kappa - \tau_\epsilon$. When $\Delta\tau > 0$, the distribution is stabilized at some arbitrarily large length $L = L_{\text{trunc}}$ by sufficiently large contraction rates in excess of all processes that increase repeat length; a truncation of the distribution (i.e., when less than one repeat is expected in a genome of

given size) occurs due to the more rapid increase of contraction rates than expansion rates that leads to contraction-biased dynamics at some point $L < L_{\text{trunc}}$. The necessity of asymptotic contraction-bias contrasts the notion that length-dependent interruptions (due to substitutions and non-motif insertions) counteract expansion at sufficiently long lengths, stabilizing the distribution[44,48,54,57–59] based on our estimated mutation rates, this effect does not lead to a steady state in the absence of contractions, as the per-repeat rate of expansions far exceeds that of repeat fission (i.e., interruptions) at long lengths. As discussed below, the length at which the contraction rate is equal to the expansion rate $L^*$ (i.e., $L^*$ is the unique length $L \geq 8$ where $\kappa_L = \epsilon_L$, which may occur at non-integer values) is highly informative about the dynamics in each regime, as well as the behavior when all effects captured in Eq. 9 are simultaneously relevant; $L^*$ is exponentially dependent on $\Delta\tau$ and more weakly controlled by the multiplier $m$, notably occurring at the same length across lines of constant $\Delta\tau$ in the parameter space (for a given $m$). For $m > 2.5$, the dynamics are nearly identical for parameter combinations with the same $\Delta\tau$, effectively collapsing the $(\tau_\epsilon, \tau_\kappa)$ plane to a single dimension. The functional dependence of $L^*$ on the parameters and further discussion is provided in the Supplementary Note.

**Effective equations approximating steady state dynamics.** Given the complexity of Eq. 9 introduced by the nonlocal effects of fission, we first searched for subsets of the contraction-biased parameter space that could be well approximated under a further reduction of the dynamics. Such simplifications are, in principle, possible because the length scaling of each term in Eq. 9 is distinct; specifically, parameter combinations exist where the nonlocal behavior (i.e., the integral representing fission in) becomes subdominant and can be neglected in our analysis. Neglecting the integral results in a second-order ODE approximation to the steady state equation. We identified two distinct dynamical regimes within the $\Delta\tau > 0$ region, which are each well-approximated by a subset of contributions that dominate the dynamics in their respective regimes of validity.

**Balance between local dynamics in the highly contraction-biased regime.** For parameter combinations with very large positive values of $\Delta\tau$ (i.e., for $\tau_\kappa \gg \tau_\epsilon$), the dynamics are entirely dominated by the diffusion and local directional flux terms appearing in Eq. 9, as the contraction rate quickly outcompetes both the rate of fission in and fission out. This results in an effective steady state equation dominated only by local transitions.

$$\frac{1}{2}\partial_L^2\left[(\epsilon_L + \kappa_L) L P_L\right] - \partial_L\left[(\epsilon_L - \kappa_L) L P_L\right] \approx 0 \tag{10}$$

In this case, the contraction rate exceeds the expansion rate almost immediately above the short length regime (i.e., $L^*$ is of order 10) such that the dynamics are effectively uniform across the long length regime. The long-length tail of the distribution decays in a super-exponential fashion such that the truncation occurs at low values of $L_{\text{trunc}} \sim 20$, which dramatically limits the lengths of repeats that occur in a genome of realistic size. In this regime, a further simplification leads to an approximate closed-form analytic solution for the rough asymptotic shape of the distribution, however, this approximation is only valid near the truncation point and rapidly loses accuracy. A more general solution was obtained by numerically solving the effective steady state equation (Eq. 10) for comparison to computational model results. To obtain numerical values, two additional constraints must be applied, as with any second-order ODE, which conceptually correspond to an overall normalization constant (in this case, fixing the relative weights of the short length and long length distributions) and a linear coefficient that defines the relative weights of two real solutions, if both exist. These constraints can be imposed

by fixing the value of the distribution at two specific lengths, $L_1$ and $L_2$, (i.e., fixing $P_{L_1} = P_{L_1}^{comp}$ and $P_{L_2} = P_{L_2}^{comp}$, where $P_L^{comp}$ is the value of the computationally modeled distribution at length $L$), with both lengths chosen to lie long length regime $L > 10$ where the continuum approximation remains valid. For consistency, we chose to constrain the numerical solutions at the two lengths of theoretical interest in stable distributions: $L_1 = L^*$ (rounded to the nearest integer) and $L_2 = L_{trunc}$, both of which definitionally remain in the long length regime at a location with finite occupancy in a realistic genome and are well defined for all values of $\Delta\tau > 0$. All numerical solutions were obtained using the *NDSolve* function in Mathematica 14.0[91]. Comparisons between computational model results and numerical solutions to Eq. 10 showed that this approximate steady state equation remains highly accurate across the $\Delta\tau \gg 1$ regime (see Supplementary Note).

**Relevant effects of fission out in the intermediate contraction-biased regime.** We found that, at less extreme values of $\Delta\tau$, roughly on the order of $\Delta\tau \sim 1$ (e.g., roughly $1.5 > \Delta\tau > 0.7$ for $m = 4$), the integral contributions to Eq. 9 remained subdominant, but the effects of fission could not be omitted completely. In this regime, fission out non-negligibly impacts the dynamics, leading to an effective steady state equation that only omits incoming contributions from fission.

$$\frac{1}{2}\partial_L^2[(\epsilon_L + \kappa_L)L\,P_L] - \partial_L[(\epsilon_L - \kappa_L)L\,P_L] - (\nu + \iota_L)L\,P_L \approx 0 \quad (11)$$

In this regime, contraction is aided by the length-reducing effects of fission out. However, the relevance of this contribution is limited roughly to lengths below $L^*$; above $L^*$, the distribution remains well-described by Eq. 10 (see Supplementary Note). This indicates that contraction is largely responsible for truncating the distribution, even when fission is involved in shaping the distribution. This defines a range of intermediate lengths below $L^*$ with distinguishable dynamics from asymptotic lengths, but this range is limited by the relatively small values of $L^*$ on the order of $L^* \sim 15-20$. The approximation in Eq. 11 is again a second-order ODE, but is complicated by the introduction of an additional length scaling associated with substitution-based fission. However, even when substitution rates are negligible (e.g., for $m \gg 1$), no exact solution could be found due to the generic power laws associated with our parameterization. For comparison to the computational model, numerical solutions were obtained by again constraining the solution at lengths $L_1 = L^*$ and $L_2 = L_{trunc}$. We found that the effective steady state equation (Eq. 11) is a highly accurate approximation to the dynamics in this regime of moderate values of $\Delta\tau$. Additionally, this approximation remains accurate at large values of $\Delta\tau$ (i.e., Eq. 11 is applicable to the full subspace $\Delta\tau \gtrsim 1$), as the approximation in Eq. 10 is nested in Eq. 11; the latter includes the additional effect of fission out, which becomes negligible for $\Delta\tau \gg 1$.

**Inclusion of the nonlocal dynamics in the weakly contraction-biased regime.** For values $\Delta\tau \lesssim 0.5$ (roughly $\Delta\tau < 0.7$ for $m = 4$), the nonlocal effects described by the integral term in Eq. 9 become relevant to the maintenance of the steady state. To further analyze this regime, we first eliminated the integral dependence by applying an overall length derivative to all terms on the right-hand side of Eq. 9, such that the equation becomes the following.

$$\partial_L\left[\frac{dP_L}{dt}\right] = 0 \approx \frac{1}{2}\partial_L^3[(\epsilon_L + \kappa_L)L\,P_L] - \partial_L^2[(\epsilon_L - \kappa_L)L\,P_L] - \partial_L[(\nu + \iota_L)L\,P_L] - 2(\nu + \iota_L)P_L \quad (12)$$

This third-order ODE now represents a constraint on the net flux, which must equal a time-independent constant. This can be seen by swapping the order of the derivatives on the left-hand side of Eq. 12: $\partial_L[dP_L/dt] = d[\partial_L P_L]/dt = 0$. Taking this overall length derivative maps the nonlocal contributions from the fission of all repeats longer

than $L$ to an effectively local boundary effect on the net flux $\partial_L P_L$ through length $L$. However, this is not equivalent to steady state until applying an additional constraint that this net flux vanishes (i.e., the special case where the constant is zero, $\partial_L P_L = 0$). Obtaining numerical solutions to this third-order ODE requires three constraints, including the constraint that the net flux vanishes. For comparison to the computational model, this was imposed by again specifying $L_1 = L^*$ and $L_2 = L_{trunc}$ along with the additional constraint $P_{L_3} = P_{L_3}^{comp}$ at length $L_3 = L_2 - 1$, chosen for convenience. We found good agreement between the resulting numerical solutions and our computational model results. Additionally, solutions to this equation accurately describe the parameter regimes that are well approximated by Eqs. 10 and 11, as the latter represent nested dynamics characterized by Eq. 12 that discard negligible contributions. Thus, Eq. 12 has a regime of validity that extends across the entire set of parameter combinations that result in stable distributions $\Delta\tau > 0$. As a corollary, the accuracy of this approximation to the full steady state dynamics across the space of computational model results indicates that the effects of repeat fusion remain negligible throughout. However, this statement is only applicable to the long repeat dynamics for $L > 10$; the effects of repeat fusion are everywhere relevant for short repeats, which, in part, shape the geometric distribution at steady state.

Details on the derivation, relevant approximations, dynamical regimes, and comparison between numerical and computational model results are provided in depth in the Supplementary Note.

## Reporting summary

Further information on research design is available in the Nature Portfolio Reporting Summary linked to this article.

## Data availability

The datasets analyzed during the current study are freely available from the NCBI, the UCSC Genome Browser (https://genome.ucsc.edu), and other studies as cited in refs. 83–90. Instructions for accessing specific datasets are further detailed in the code repository (see "Code availability"). DRLs for mammalian genomes analyzed in this study are provided in Supplementary Data 1. Length-dependent instability rates calculated in this study are provided in Supplementary Data 2.

## Code availability

The code to perform the analysis in the current study is available in a GitHub repository (https://github.com/ryanmcggg/repeat_distributions)[92]. Software/packages (including version numbers) are further detailed therein.

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

## Acknowledgements

This study was supported by grants from NIGMS R35GM127131 (S.R.S.), NIMH R01MH101244 (S.R.S.), NHGRI U01HG012009 (S.R.S.), NIGMS R35GM130322 (S.M.M.), and NSF-BSF 2153071 (S.M.M.). Portions of this research were conducted on the O2 High Performance Compute Cluster, supported by the Research Computing Group, at Harvard Medical School. We thank Alexey Kondrashov and Alisa Lyskova for helpful discussions at the early stages of the project.

## Author contributions

R.J.M. conceived the study and performed bioinformatic analyses. R.J.M. and D.J.B. jointly designed and implemented the computational model and prepared the manuscript. D.J.B. and SS.R. constructed the analytic

model and mathematical analyses. R.J.M., D.J.B., and S.R.S. devised the statistical inference procedure. S.M.M. and S.R.S. supervised the project, discussed results, and prepared the manuscript.

## Competing interests

The authors declare no competing interests.
