## [Transparent Peer Review file · Nature Communications]

Inherent instability of simple DNA repeats shapes an evolutionarily stable distribution of repeat lengths

Corresponding Author: Dr Shamil Sunyaev

Version 0:

Reviewer comments:

Reviewer #1

(Remarks to the Author)

The paper by McGinty et al. studies the length statistics of simple repeat sequences in mammalian genomes. The authors observe well-conserved length distributions with a broad, approximately power-law tail and analyze the emergence of such distributions from mutation models. A key findings is that the shape of the observed distributions can be explained by an expansion-contraction-mutation dynamics alone, without invoking natural selection. In particular, they argue that the neutral fission dynamics is sufficient to constrain the length of most repeats below a level associated with disruption of gene function and disease. Further, the authors find evidence of length-dependent insertion rates indicative of a repeat instability regime setting on at lengths above 10nt.

The paper is, overall, a careful and timely study of this topic and presents valuable results. However, some of the statistical methodology and some aspects of presentation of the results are not yet convincing and should be improved in a revision. Specific comments:

1. The class of models chosen to parametrize the data is relatively complex. The main feature of the observed distributions $P(L)$ is the presence of long-tails with an approximately power-law form. There is clearly some additional structure in these distributions, i.e., deviations from straight lines in the log-log plots of Fig. 1. However, you would need to show that this structure gives statistically significant evidence for the model chosen, compared to simpler models.

It is known that expansion-mutation models with constant rates can give generic power-law solutions of stationary states that depends on a single effective parameter (probably with additional fission effects for the length distribution at large L). See, e.g. Messer et al. Physical. Rev. Lett. 2005, which contains a simpler version of your equation (3). Choosing appropriate constant-rate dynamics with power-law solutions as null model, does the full model explain the data significantly better? Perhaps you could show a table that compares the log-likelihood scores of the full model with subleading models of decreasing complexity and with different functional forms of the length-dependent rates, as well as a modified Figure 3 that compares different models with the data. Specifically, you could then argue that the data harbor clear statistical evidence for the transition to instability at a given length threshold, which is important for the biological conclusion but currently lacking in the manuscript.

2. I understand your reasons for picking the metric of equation (1) for comparing different forms of $P(L)$, namely, to have a measure sensitive to differences in the tail. However, this metric looks a bit ad hoc and not related to a likelihood score in a straightforward way. In view of the expansion dynamics underlying $P(L)$, perhaps a better option could be $\sum_L \log \text{Prob}(N(L + \ell) | N(L))$ or so, where $N(L)$ denotes again the counts (unnormalized distribution) and the probability is evaluated under a given model. Such a measure would look at the deviation in slope of the measured log log curve $P(L)$ and a model.

Importantly, the model inference and likelihood analysis should be carried out on the ensemble of species-dependent patterns $P(L)$ and not on their median.

3. The presentation of the paper is somewhat convoluted, making the reading more tedious than necessary. There are too many bioinformatic details in the main text; at the same time, one has to search for a clear statement of functional forms, e.g., $\mu_{\text{insertion}} = A L^{-\tau_e}$ for $L > L_0$. It would be great to include a schematic Figure 1 with all relevant processes and their rates.

4. I suggest to give actual number densities or cumulative numbers instead of normalized distributions for the genomic data (current Fig. 1); readers would like to get an idea of the actual numbers to judge the underlying stochasticities at large L in the reported genomic data.

(Remarks on code availability)

Reviewer #2

(Remarks to the Author)

Instability of short repetitive sequences causes numerous human disorders. McGinty and colleagues here analyze the distribution and evolution of repetitive DNA sequences, with the aim to provide insights into their stability and prevalence, and compare that to random distribution and mutagenesis. While I think this is a very worthwhile effort, as similar studies from the genomic era do not exist, and such report could thus be a very important resource for the community. At its present form however, the study seems premature and difficult to access.

- More effort should be given to improve the readability of the whole text. While I understand that most biologists will not understand the modeling no matter what, the authors should try to make at least the conclusions more accessible.
- In accord with the point above, data presentation often lacks clarity, it is not clear why particular datasets were chosen, and proper explanation is often not given in the text/legends.
- To my understanding most pathogenic repeats are rather short (most of them being trinucleotides, with some tetra/penta-repeats). The authors, in multiple contexts, come to the conclusions that repeats above 10 nt or so behave differently than shorter ones. The relevance of these observations is not clear.
- The authors could try to compare repeat distribution/prevalence with a calculated susceptibility to form secondary DNA structures.

Other specific comments:

Fig. 1a: please define how unit length differs from repeat length; check X axis labels (should be e.g. 10, 20, 50, 100, correct?). In the legends, make more clear how real genome data vs. calculated are presented.

If I understand the data correctly, Fig. 1 shows that repeats shorter than 10 nt are roughly randomly distributed; e.g. CAG repeat is not more present than shuffled variation of these bases. Is that correct? To which extent is this analysis affected by codon usage? Do elements from the LINE/SINE sequences cause the bias of the longer repeat length?

Figure S1 the legend could benefit from a brief explanation of ‘-“non-normalized” versus “normalized” data and what that means

Fig. 2a shows that repeat instability increases with repeat length, with a decrease after 10 nt. It is not clear why this is not seen in Fig. S3. How were the repeats in S3 chosen? For the rapid change in instability for AAGG repeat between more or less 7-8 units, do the authors suggest that there is a secondary structure formation at the cutoff? The most common pathogenic repeat are repeats including CAG, CGG, GAA, CTG, ATTCT - why were they not included in the analysis?

The authors mention that the long repeat tail includes alleles near disease threshold. It would have been helpful if the discussion elaborated on how their model might be applied or extended to understand the pathological expansions seen in disorders like Huntington's syndrome.

The authors rely on counting interrupted tandem repeats and treating interruptions as breaks that split repeats into independent units. While this is supported by previous studies, the discussion could acknowledge potential nuances. For instance, in some contexts, interruptions might stabilize a repeat tract. Discussing this possibility in more detail (and how it might influence the steady state) would add nuance to the claims about selection versus mutation.

Figure S5,6,8: adding an inset or schematic that explains, in lay terms the meaning of each parameter in the computational model (such as m , τ_e , and τ_k) could make these figures more accessible.

(Remarks on code availability)

Reviewer #3

(Remarks to the Author)

This is a very interesting manuscript arguing that repeat length distributions in hominoids and other mammals can all be explained by a unified mutational process involving expansion, contraction, and accelerated mutagenesis at long repeat length scales. I found the manuscript to be extremely careful, well explained, and informative, and I have no major complaints. The one thing I would suggest the authors revisit is the potential role of bioinformatic errors, which are not

discussed much and can be significant for this kind of variation. In particular, homopolymers repeats are known to be especially error-prone--could these errors explain any of the difference observed between homopolymer repeats and higher order repeats? There are large differences in estimated mutation rates between some recent papers on de novo STR mutations, e.g. Mitra, et al. 2021 doi:10.1038/s41586-020-03078-7 estimated STR mutation rates that were quite a bit higher across the board than some other studies such as Goldberg, et al. 2024 <https://doi.org/10.1093/genetics/iyae013>. Can this model speak to such differences in DNM rate estimates and which algorithms for repeat DNM calling appear to be most accurate? Conversely, how much of the parameter inferences from this paper depend on precision of the mutation rate estimates in the input trio data?

(Remarks on code availability)

Version 1:

Reviewer comments:

Reviewer #1

(Remarks to the Author)

Thanks to the authors for a careful revision that addresses the points raised in my previous report. Specifically, the statistical model comparison is now more robust and an appropriate null model is introduced. The overall presentation is still quite detailed but structured more clearly than in the original manuscript. The present form appears ready for publication.

(Remarks on code availability)

Reviewer #2

(Remarks to the Author)

I thank the authors for answering my questions. I also apologize if I did not understand some points. The authors certainly made effort to make the study more accessible. The study will certainly be informative to researchers interested in repeat evolution and distribution across genomes.

(Remarks on code availability)

Reviewer #3

(Remarks to the Author)

The authors have done an excellent job with revisions. I have no further comments.

(Remarks on code availability)

The GitHub repository looks superficially very complete and well organized. I did not review the code in detail.

We thank the reviewers for their careful consideration of our manuscript. Based on the many helpful comments provided in these reviews, we have substantially revised our manuscript to clarify the text, make our statistical analysis more robust, and incorporate nearly all suggested changes. A point-by-point response to each reviewer is provided below, with the original review in blue and our responses in black.

REVIEWER COMMENTS

Reviewer #1 (Remarks to the Author):

The paper by McGinty et al. studies the length statistics of simple repeat sequences in mammalian genomes. The authors observe well-conserved length distributions with a broad, approximately power-law tail and analyze the emergence of such distributions from mutation models. A key findings is that the shape of the observed distributions can be explained by an expansion-contraction-mutation dynamics alone, without invoking natural selection. In particular, they argue that the neutral fission dynamics is sufficient to constrain the length of most repeats below a level associated with disruption of gene function and disease. Further, the authors find evidence of length-dependent insertion rates indicative of a repeat instability regime setting on at lengths above 10nt.

The paper is, overall, a careful and timely study of this topic and presents valuable results. However, some of the statistical methodology and some aspects of presentation of the results are not yet convincing and should be improved in a revision.

We thank the reviewer for their very detailed reading of our manuscript and for a particularly insightful critique. We took the overall recommendation to revise the statistical procedure quite seriously and completely revamped our inference methodology to now provide Approximate Bayesian Computation-based posterior distributions for a number of parameterizations. We hope the reviewer will find the statistics substantially more robust than in our initial submission. Following the reviewer's suggestions, we believe the overall presentation is now improved in its clarity and hope that the argument is more persuasive.

Specific comments:

1. The class of models chosen to parametrize the data is relatively complex. The main feature of the observed distributions $P(L)$ is the presence of long-tails with an approximately power-law form. There is clearly some additional structure in these distributions, i.e., deviations from straight lines in the log-log plots of Fig. 1. However, you would need to show that this structure gives statistically significant evidence for the model chosen, compared to simpler models.

We thank the reviewer for these comments and have now implemented a completely revised inference framework using an Approximate Bayesian Computation (ABC) procedure (following the methodology in Wilkinson (2013) [1]), which is central to our revised analysis. Doing so allowed us to compare various parameterizations via their Bayes Factor Ratio to assess the relative statistical support for each parametric model

of the length-dependent instability rates (these comparisons are now summarized in **Table 1** and referred to throughout the Results section of the manuscript).

To address the reviewer's concern that the model specified in our first submission is overcomplicated, we now present results for a series of nested models (i.e., parameterizations) and compare their Bayes factors to assess their relative statistical support. At the reviewer's suggestion, we now include an appropriate two-parameter 'null model' based on relevant literature, which specifies equal expansion and contraction rates (i.e., no expansion-contraction bias) at all lengths. This parameterization represents an oversimplified description of replication slippage, which is appealed to as the major biological mechanism responsible for rapidly increasing rates with increasing length in numerous previous studies (e.g., the model described in Kruglyak, et al. (1998) [2]). We feel this provides the most appropriate null for the mutational model and can be used for comparison to parameterizations with additional degrees of freedom (e.g., representing expansion-contraction bias). After comparing Bayes Factors across several parameterizations, we found that this null model is statistically disfavored relative to parameterizations that decouple the expansion and contraction rates (to varying degrees) because the resulting distributions of repeat tract lengths (DRLs) do not resemble empirical distributions.

It is known that expansion-mutation models with constant rates can give generic power-law solutions of stationary states that depends on a single effective parameter (probably with additional fission effects for the length distribution at large L). See, e.g. Messer et al. *Physical. Rev. Lett.* 2005, which contains a simpler version of your equation (3). Choosing appropriate constant-rate dynamics with power-law solutions as null model, does the full model explain the data significantly better?

We thank the reviewer for this reference, which presents an elegant analysis of related phenomena. We now cite Messer, et al. (2005) [3] in the context of previous theoretical descriptions of repeat length dynamics. Two important distinctions exist between the author's analysis and the present work. First, the authors model a simplified version of the expansion-contraction-insertion-substitution dynamics in the absence of explicit length dependent rates (i.e., rates per target are constant, and rates per 'repeat' scale linearly with target size). While this is qualitatively inconsistent with both our empirical estimates and previous experimental results that show strong monotonic length-dependence of per target instability rates, this toy model allows the authors to explicitly solve for the correlation between bases in closed form. The second distinction is in the primary object representing genome-wide properties of repetitive sequences. Instead of focusing on the correlation between base content, we assess the dynamical behavior of the distribution of repeat lengths, treating an *uninterrupted* repeat as the fundamental object. While these quantities are related, correlations between bases allow for non-contiguous (i.e., interrupted) sequences and must be summed over all repeat lengths. As a result, the power laws presented in Messer, et al. are not immediately comparable to power law-like behavior of the distribution of uninterrupted repeat lengths. In particular, our Equation 3 characterizes the set of length transitions that alter uninterrupted repeat length, while their manuscript details the master equation for the

sequence correlation. Other distinctions exist, precluding direct use as a null model, but their work is clearly related and of interest to the questions addressed in our manuscript.

As a result of the reviewer's suggestions, we now present a model comparison based on a biologically motivated null model (discussed above). Additionally, our full parameter space includes parameter values that model asymptotic dynamics under the same assumptions made in Messer, et al., with constant, expansion-biased instability rates that lead to the power law dependence in the correlation function shown in their manuscript. Such models are rejected in favor of more rapid power-law growth in both the expansion and contraction rates; the prior distribution we now use to incorporate popSTR-based instability rate estimates suppresses the posterior probability for all constant rate models of instability (these models are also qualitatively inconsistent with experimental data showing rapidly increasing rates). Having said that, the steady-state distribution resulting from asymptotically constant models is informative about the distinction between the two observables; while the correlation between sites decays as a power law, the distribution of uninterrupted repeats decays more quickly due to the increased rate of fission at longer lengths (i.e., the presence of interruptions modifies the equilibrium distribution in an intuitive way, more severely depleting each bin as length increases).

Perhaps you could show a table that compares the log-likelihood scores of the full model with subleading models of decreasing complexity and with different functional forms of the length-dependent rates, as well as a modified Figure 3 that compares different models with the data.

We thank the reviewer for this extremely helpful suggestion, which largely motivated our change of inference procedure. Unfortunately, directly computing a realistic likelihood proved intractable, so we instead relied on ABC to estimate posterior probabilities without an explicit likelihood. Within this Bayesian framework, Bayes factor ratios can be used for model comparison, analogous to performing likelihood ratio tests. We now use a nested set of power-law-based models to identify the parameterization with the strongest statistical support. Thankfully, the multiplier-based parameterization described in our initial submission is best supported such that our understanding of the dynamics remains the same. In addition, we tested two distinct functional forms (asymptotic power law and powers of less extreme logarithmic growth) relative to their respective 2-parameter null models, both of which rejected the latter in favor of including expansion-contraction bias parameters. Because we lacked explicit likelihoods to construct truly uninformative Jeffrey's priors, comparison across functional forms is complicated by the differences in parameter space. While these functional forms are likely dramatically oversimplified descriptions of the true length dependences, the qualitative results remain the same for both. These parameterizations are now defined in Methods; model comparison results are now described in the main text and summarized in **Table 1**.

We again thank the reviewer for this suggestion, which we feel substantially improved the statistical support for our claims and the manuscript, in general.

Specifically, you could then argue that the data harbor clear statistical evidence for the transition to instability at a given length threshold, which is important for the biological conclusion but currently lacking in the manuscript.

First, we note that a rough onset length for repeat instability can be observed directly in the estimated rates of repeat instability: at some intermediate length, one or both instability-related rates become comparable to (and eventually dominate over) the substitution rates. This necessarily perturbs the steady-state distribution from substitution-driven geometric decay by altering the rate of local transitions in length. That said, we again thank the reviewer for suggesting a more robust analysis that we hope will help bolster our conclusions. We have now introduced two additional related analyses in the manuscript Results section.

We first performed a rate estimate-free analysis to characterize the qualitative information contained in the shape of the DRL. We defined a ‘pure power-law’ parameterization expressed in terms of the exponent and the length at which the inferred rate intersects the estimated substitution rates (specifically, μ , the lower of the two substitution rates, which is intersected first). This intersection length serves as one way to define and infer the onset length of repeat instability from the steady state distribution without appealing to rate estimate-based observations, which are perhaps more susceptible to statistical and systematic noise, especially for homopolymer repeats. The absence of empirical rate estimates (otherwise used to explicitly fix instability rates for $L < 9$ and define an informative prior) ensures that any inferred onset length is independent of systematic errors or intersections observable in our direct rate estimates. This is now included as the penultimate subsection in Results (with the header: ‘*Inferring the onset of instability from the shape of the DRL*’), along with a corresponding supplementary figure (**Figure S13**) showing the marginal distribution for intersection lengths inferred from the DRL alone. These marginal distributions are localized to a single onset length at $L=9$, (the lower of the two inferred intersection lengths, which captures the initial deviation from a geometric steady state distribution), which is roughly at the length observed in our direct expansion and contraction rate estimates in mono-A repeats. Further, this demonstrates that the DRL can be used to constrain the instability onset length, provided an appropriate mutational model is used for inference.

Second, we performed a heuristic qualitative analysis to compare the rough onset length independently assessed from rate estimates and from the DRL for various motifs. This now appears in the last subsection of Results (with the header ‘*Application to longer motif unit lengths*’) and in **Figure S14**. While this analysis is largely qualitative, we hope that 1) the general agreement between these independent observations and 2) the constrained onset length range across motifs (roughly at 6-12 nt for nearly all motifs) will help to justify our conclusions about the universality of the transition to repeat instability.

2. I understand your reasons for picking the metric of equation (1) for comparing different forms of $P(L)$, namely, to have a measure sensitive to differences in the tail. However, this metric looks a bit ad hoc and not related to a likelihood score in a straightforward way. In view of the expansion dynamics underlying $P(L)$, perhaps a

better option could be $\sum_L \log \text{Prob}(N(L + \ell_0) | N(L))$ or so, where $N(L)$ denotes again the counts (unnormalized distribution) and the probability is evaluated under a given model. Such a measure would look at the deviation in slope of the measured log log curve $P(L)$ and a model.

We appreciate the reviewer's suggestion and agree that our original choice of metric was somewhat specific to our previous analysis. As the true likelihood is analytically insoluble, our revised Bayesian inference is performed by Approximate Bayesian Computation, a well-established procedure that approximates the posterior distribution by relying on a summary statistic in lieu of specifying an explicit likelihood. We used the Kullback-Leibler divergence (a commonly used statistic in applications of ABC) to summarize the divergence between modeled and empirical DRLs (see **Equation 1** in Methods). Our implementation of ABC follows the prescription detailed in Wilkinson (2013) [1] (see Methods). We hope that use of this more standard inference procedure addresses concerns related to the ad hoc metric specified in our previous analysis.

Importantly, the model inference and likelihood analysis should be carried out on the ensemble of species-dependent patterns $P(L)$ and not on their median.

We again thank the reviewer for this comment and have now defined the rejection threshold for the KL divergence in our ABC inference procedure based on the ensemble of values between each primate and the human reference; this is performed with a soft rejection probability that is Gaussian suppressed in the KL divergence with a variance defined by the approximate range of primate divergences (after removing outliers). In this way, the posterior probability distribution over parameter space is explicitly constructed around the observed distribution of primate DRLs (see '*Bayesian inference procedure*' in Methods).

3. The presentation of the paper is somewhat convoluted, making the reading more tedious than necessary. There are too many bioinformatic details in the main text; at the same time, one has to search for a clear statement of functional forms, e.g., $\mu_{\text{insertion}} = A L^{-\tau_e}$ for $L > L_0$. It would be great to include a schematic Figure 1 with all relevant processes and their rates.

We apologize for any confusion caused by the presentation in our original manuscript. Due to the mixed readership that includes repeat biologists (part of the intended audience for our work), we have decided to maintain some of the bioinformatic details in Results that constitute new observations central to the arguments made in the manuscript. However, at the reviewer's suggestion, we have substantially rewritten the text to improve clarity in several ways. We moved some of the bioinformatic details to Methods and more clearly isolated remaining bioinformatic results by restructuring our Results into more appropriate subsections (e.g., bioinformatic results associated with the DRL are now within the subsection '*Features of the repeat length distribution and evolutionary stability*'; direct instability rate estimate results at low and intermediate tract lengths are detailed in the subsection '*Mutational transitions in repeat tract length*').

These subsections are separated from inference and analytic results, which we hope will improve readability of the manuscript. Additionally, at the reviewer's suggestion, we have added **Table 1** to summarize both the functional form for each parameterization and the associated Bayesian inference results; the parameterizations are also now defined in **Equations 5–7** in Methods (in the subsection '*Parameterizations of repeat instability rates*') for clarity.

In addition to substantial restructuring of the text, we have now added two schematic figures presenting: 1) enumeration of mutational processes and the appropriate maps to length transitions; and 2) a flow diagram detailing the Bayesian inference procedure, which is somewhat involved. We hope that these visual summaries of both the fundamental processes and the inference pipeline will allow readers to more easily follow our analyses.

4. I suggest to give actual number densities or cumulative numbers instead of normalized distributions for the genomic data (current Fig. 1); readers would like to get an idea of the actual numbers to judge the underlying stochasticities at large L in the reported genomic data.

Figure 1a includes non-normalized distributions (summed over motifs of the same length); **Figure 1b** shows the normalized DRLs constructed from various mammalian genome assemblies. While normalization is clearly necessary to compare assemblies of distinct lengths (both across species and sequencing technologies), we agree that the raw counts are informative about the threshold to statistical noise (and potentially stochastic temporal dynamics). To improve communication of this property (i.e., the truncation length for each assembly), we now present the non-normalized mammalian DRLs for mononucleotide-A repeats (as in **Figure 1b**) alongside the normalized comparison in **Figure S1**; we have also added violin plots to the same figure that show the distribution of assembly lengths across mammals and primates. At the request of another reviewer, we have also added a section to Methods that better explains the rationale and provides details related to normalization of the DRLs to facilitate our inference of the instability rates.

Reviewer #2 (Remarks to the Author):

Instability of short repetitive sequences causes numerous human disorders. McGinty and colleagues here analyze the distribution and evolution of repetitive DNA sequences, with the aim to provide insights into their stability and prevalence, and compare that to random distribution and mutagenesis. While I think this is a very worthwhile effort, as similar studies from the genomic era do not exist, and such report could thus be a very important resource for the community. At its present form however, the study seems premature and difficult to access.

- More effort should be given to improve the readability of the whole text. While I understand that most biologists will not understand the modeling no matter what, the authors should try to make at least the conclusions more accessible.
- In accord with the point above, data presentation often lacks clarity, it is not clear why

particular datasets were chosen, and proper explanation is often not given in the text/legends.

We have undertaken extensive revisions to improve clarity throughout the text and figures. We hope that the reviewer will find the new text more comprehensible. Due to the mixed readership, we have now restructured the Results by separating bioinformatic results in appropriate subsections that we hope will be more accessible to more biological readers. Similarly, more technical aspects of our analysis are now more self-contained in subsections. We hope that our revised Discussion is more generally accessible by focusing on qualitative insights, rather than technical details.

Regarding the choice of datasets, the Introduction includes the following statement that motivated our analysis:

“Technological developments led to the release of the human Telomere-to-telomere genome (T2T-CHM13), which more than doubled the number of mapped simple repeats compared to the previous reference genome GRCH381. This warranted a fresh look at the distribution of repeat lengths and whether mutational processes, in the absence of selection, can explain their abundance.”

At the reviewer’s request, we have added justification for the choice of datasets used to produce instability rates estimates. The Methods section on “*Bioinformatic estimation of substitution and indel rates*” now includes the following text:

“De novo mutation datasets were acquired as VCF files (or equivalent) from various published sources⁷⁵⁻⁸², representing a total of 10,912 parent-child trios with available SNV data and 9,387 trios with available indel data. This dataset was compiled in McGinty and Sunyaev (2023) and comprised of all freely available trio samples at the time of analysis; samples from distinct VCFs were pooled to increase statistical power.”

“The popSTR repeat instability dataset, representing 6,084 parent-child trios, was acquired from the supplement of Kristmundsdóttir, et al. (2017). This dataset was incorporated into our inference due to the unique methodology, which provided high quality calls of mutations extending beyond short tract length repeats that allowed us to produce length-stratified rate estimates.”

To my understanding most pathogenic repeats are rather short (most of them being trinucleotides, with some tetra/penta-repeats). The authors, in multiple contexts, come to the conclusions that repeats above 10 nt or so behave differently than shorter ones. The relevance of these observations is not clear.

We believe the reviewer is conflating *repeat unit length* with *repeat tract length*. Indeed, most diseases are associated with trinucleotide motifs (i.e. unit length = 3), such as (CAG)_n in Huntington’s Disease. Here, *L* describes the tract length of the repeat in multiples of the unit length, which, to use the same example, causes disease roughly when $L \geq 30$ repeated units. It is therefore important to understand the behavior of long repeat tracts. We observe that repeat instability, the mutational mechanism underlying

disease progression, begins to dominate (over the background substitution rate) when tract length exceeds ~10 nt, regardless of the motif or the motif unit length. Understanding the tract length at which this transition occurs (from repeat-independent mutation to repeat instability) is thus important to unraveling the underlying molecular mechanisms. Because the distinction between unit length and tract length is critical to the interpretation of our results, we have modified the text to make this distinction clear by consistently referring to 'unit length' or 'tract length' throughout the manuscript to avoid confusion.

The authors could try to compare repeat distribution/prevalence with a calculated susceptibility to form secondary DNA structures.

One of our key observations is that every simple tandem repeat motif has a repeat length distribution with a tail of long repeats, indicating that the dominant mechanism(s) responsible for this excess of longer repeat tracts acts in a secondary-structure-independent manner; the focus of this study is on understanding this universal nature of this observation. We do observe some amount of variation between the distributions for different repeated motifs, which indeed may be attributable to differences in secondary structure formation. While we appreciate the suggestion to compare fine-scale distinctions between motifs with a measure of secondary-structure stability (e.g., free energy), this is beyond the scope of the current study and would require a substantial effort worthy of its own manuscript.

Other specific comments:

Fig. 1a: please define how unit length differs from repeat length; check X axis labels (should be e.g. 10, 20, 50, 100, correct?).

Please see our response above, regarding the difference between repeat unit length and repeat tract length. To avoid confusion, we have now standardized the x-axis labels in all relevant figures to denote the repeat tract length (referring to number of repeated units or total number of nucleotides, where appropriate).

In the legends, make more clear how real genome data vs. calculated are presented.

We have updated our figure legends to ensure this distinction is clear.

If I understand the data correctly, Fig. 1 shows that repeats shorter than 10 nt are roughly randomly distributed; e.g. CAG repeat is not more present than shuffled variation of these bases. Is that correct?

We believe the reviewer may again be conflating repeat unit length with repeat tract length. The reviewer may be viewing the x-axis as unit length, rather than tract length, leading to the misinterpretation that CAG repeats (unit length = 3) are present in the same numbers in both the T2T genome and the shuffled genome sequence. Rather, CAG repeats are included in the curve labeled "motif unit length = 3" which pools over all trinucleotide motifs. The plot demonstrates that, for repeat tract lengths of less than

roughly 10 nt, the number of repeats in the reference sequence is approximately the same as the number counted after shuffling. Focusing on unit length = 3 as an example: 6 nt repeats (e.g. (CAG)₂ = CAGCAG) appear to be assembled randomly; 9 nt repeat tracts, (CAG)₃, deviate slightly from the shuffled sequence; and tract length >12 nt repeats are clearly overrepresented. We hope that our revisions have clarified this distinction.

To which extent is this analysis affected by codon usage? Do elements from the LINE/SINE sequences cause the bias of the longer repeat length?

The relevance of codon usage to the repeat tract length distribution within coding regions of the genome is indeed of interest. However, codon usage does not meaningfully alter the genome-wide distribution due to the small target size of coding sites relative to the remainder of the genome (~2%). While specific focus on coding sequences is both warranted and relevant to natural selection, such an analysis lies beyond the scope of the present manuscript, which focuses on establishing a purely mutation-based null model of the genome-wide behavior. An adequate treatment of this topic would likely require substantially more specific biological information, including the relevance of protein and RNA structure, the biological roles of amino acid repeats, a robust estimate of coding-specific mutation rates, and other coding-specific details. We thank the reviewer for highlighting this topic, which we hope will become a subject of follow-up research.

We note that we do have preliminary results for sub-compartments of the genome, including LINE/SINE subsets and restricting to coding sites. However, these analyses involve additional bioinformatic (e.g., dependence of mutation rate estimation on unique mapping of mobile elements via long read population data), statistical (e.g., power limitations on smaller target sizes), and evolutionary complications (e.g., younger categories like SVAs, which are hominid-specific, likely have yet to equilibrate) that still need to be addressed prior to presenting associated results. We agree with the reviewer that this is a particularly interesting line of questions and hope to address these topics in a follow-up publication with more attention on variation across genomic categories.

Figure S1 the legend could benefit from a brief explanation of ‘-“non-normalized” versus “normalized” data and what that means

Due to distinct genome lengths (i.e., the number of sequenced bases in a given genome assembly), inter-species (e.g., **Figures 1b, S2**) and cross-technology (e.g., **Figure S1b**) distributions are directly comparable only after normalization. We believe that the concept of normalization (i.e., dividing by the sum of a distribution to create a fraction of the distribution) is well understood by the wider scientific audience. However, we have added a more detailed description of this procedure to a new Methods subsection called “*Distribution normalization*”. We have also added a graphical depiction of the normalization process as new Supplementary Figures S1 d and e.

Fig. 2a shows that repeat instability increases with repeat length, with a decrease after 10 nt. It is not clear why this is not seen in Fig. S3.

In our initial submission, **Figure 2a** was identical to that in the first panel in **Figure S3** showing mononucleotide-A repeats (the same figures are now **Figure 3b** and **Figure S4a** in our revised manuscript). However, the reviewer may be asking about the variation observed between the various motifs depicted in **Figure S3**. Decreases in the rates estimated from pooled trio data for tracts above 10 nt can indeed be seen for many motifs, though not all. Because of the large number of distinct motifs represented in **Figure S3**, each plot is small and may be difficult to parse visually (especially the difference between the pooled trio and popSTR datasets). To remedy this, we have now included a combined plot (**Figure S4b**) where the expansion and contraction rates are shown after pooling all motifs of the same unit length. In these larger plots, similarities between instability rates for differing unit lengths are more visually apparent.

As the reviewer notes, instability rate estimates at longer tract lengths show an apparent decrease in some motifs. This occurs at tract lengths where repetitive sequences generated from short read data are unreliable (and are called in low numbers, generating large statistical noise). Rate estimates at these lengths are included in our manuscript to show the systematic decrease in quality for long repetitive tracts. As these estimates are unreliable and subject to large systematic errors, they were not included in our analyses or inference. In our revised manuscript, we have removed estimates we know to be systematically biased from our main figures to avoid confusion; discussion of these errors now appears in the Methods subsection titled “*Bioinformatic estimation of substitution and indel rates*” and included in supplementary figures. While these technical details may be of interest to bioinformaticians, our focus is on the subset of reliable rate estimates relevant to the dynamics of the distribution of repeat tract lengths.

How were the repeats in S3 chosen?

Figure S3 (formerly; now **Figure S4a**) was intended to comprehensively display all repeat motifs with unit lengths of 1 to 4 nt (see below for explanation of redundant motif labeling); however, a typo led to several motifs being omitted or placed out of alphabetical order. We thank the reviewer for pointing this out and have corrected this issue.

Figure S4a now shows all irreducible repeat motifs with unit length below 5 nt. Repeats are pooled over cyclic permutations (e.g., ‘ACG’ labeled counts include ACG, GAC and CGA repeats, which are indistinguishable up to a single unit on either end, and their reverse complements CGT, GTC and TCG). Labels for each category correspond to the first permutation in alphabetical order. Pooling over equivalent motifs increases our statistical power but associated results are insensitive to genomic features that can break directional symmetry such as gene direction or coding frame; this pooled data is most appropriate for analysis of the genome-wide behavior of repeats. Counting over permutations is now detailed in the Methods section under the subheading “*Motif labeling*” and in **Figure S1c**. Consequently, no choice was made when presenting this data, except to omit estimates for longer unit lengths; low counts and combinatoric growth in the number of motifs (e.g., there are over 100 motifs with

unit length 5) results in both poor estimation and prolific enumeration for unit lengths above 4.

For the rapid change in instability for AAGG repeat between more or less 7-8 units, do the authors suggest that there is a secondary structure formation at the cutoff?

Again, we apologize for the small size of the plots in **Figure S3** (formerly, now **Figure S4a**). Note that the separate pooled trio and popSTR datasets are represented by squares and circles, respectively. This should be more visually apparent in the larger plots in the new **Figure S4b**. The “rapid change” at ~8 units that the reviewer is observing is due to pooled trio data ending and popSTR data beginning at this length. As we now describe in the Methods section on “*Bioinformatic estimation of substitution and indel rates*”, non-trivial differences in these two datasets prevents direct comparison between estimated instability rates, especially at the boundary between their respective length ranges that include reliable rate estimates. The reviewer’s observation is likely due to artefacts precluding direct comparison across these datasets, rather than owing to secondary structure. However, the following text in our Discussion describes secondary structure as a possible source of variation between the distributions of different motifs:

“Slipped-strand structures may be a motif-independent source of loop-outs subject to the same MMR-processing; in contrast, other secondary structures are motif-specific and therefore cannot be the primary source of repeat instability but can potentially explain differences between motifs⁶⁴ (**Figure S1a**).”

The most common pathogenic repeat are repeats including CAG, CGG, GAA, CTG, ATTCT - why were they not included in the analysis?

Please see the above reply on pooling over motif permutations. CAG and CTG motifs are combined with AGC, GCA, TGC and GCT motifs, all of which are included under the (alphabetically-first) label, ‘AGC’. GAA is included under ‘AAG’; CGG is included under ‘CCG’. We have clarified this in Methods (see ‘Motif labeling’) and in the figure legends, where appropriate. We have also added an inset **Figure S1c** to highlight examples for the labeling of several well-known pathogenic repeat motifs.

The authors mention that the long repeat tail includes alleles near disease threshold. It would have been helpful if the discussion elaborated on how their model might be applied or extended to understand the pathological expansions seen in disorders like Huntington’s syndrome.

Indeed, this is a very interesting extension of the present study. While our immediate goal in the present work is to establish the aggregate behavior of the genomic landscape of repeats, this is motivated, in part, by the need to establish a baseline mutagenic model for disease loci. Additionally, during this review process, another study was published by Handsaker, et al. (2025) [4], which explores the length range at and above the disease threshold length. Interestingly, while we find that sufficient

contraction bias at sub-disease lengths is essential for maintaining the long term stability of the genome-wide repeat length distribution, Handsaker, et al. find that an expansion bias is required to produce the extremely long repeats observed somatically in patients. We have added a paragraph to the Discussion (third paragraph in our revised manuscript) that directly addresses this apparent contradiction. We hope to address the transition from long-normal repeat length to the disease length threshold, along with related topics, in future analyses.

The authors rely on counting interrupted tandem repeats and treating interruptions as breaks that split repeats into independent units. While this is supported by previous studies, the discussion could acknowledge potential nuances. For instance, in some contexts, interruptions might stabilize a repeat tract. Discussing this possibility in more detail (and how it might influence the steady state) would add nuance to the claims about selection versus mutation.

As noted, previous studies indicate that the instability of the locus (consisting of two or more tracts split up by interruptions) relates to the longest uninterrupted repeat tract. Interestingly, the fact that interruptions stabilize a repeat locus is a feature that naturally emerges from our model. By splitting a repeat into two independent and shorter repeats, the rate of expansion of each portion is reduced by a fold-decrease dependent on the position of the interruption. This is consistent with a recent pre-print by Hujoel, et al. (2024) [5] that analyzed interruptions under this framing. We now note this consistency with previous studies in the Results section:

“This treatment of interruptions as effectively splitting a repeat is consistent with previous observations that interruptions result in locus-wide rates scaling with the longest contiguous subunit or, alternatively, a rate reduction that scales with distance from the repeat boundary.”

Because of the rapid increase in rates with increasing repeat tract length, the sum of the rates of the two independent repeats is less than the rate of the single uninterrupted motif, often by orders of magnitude. Reframing this in terms of ‘repeat fission’ therefore offers a natural explanation for past descriptions of stabilization via interruption. We also note the small, yet non-negligible role for interruptions in shaping the steady-state shape of the genome-wide distribution of repeat tract lengths. We thank the reviewer for the suggestion to emphasize this aspect of the model.

Figure S5,6,8: adding an inset or schematic that explains, in lay terms the meaning of each parameter in the computational model (such as m , τ_e , and τ_k) could made these figures more accessible.

We thank the reviewer for this suggestion. Along these lines, we have added two schematic figures to our revised manuscript that we hope will make our analysis easier to follow. **Figure 2** shows a visual representation of the relevant mutational processes (e.g., expansion rate ϵ , contraction rate, κ , etc.) and how this relates to transitions in the length distribution (e.g., length increases and decreases, fission, etc.). Separately,

Figure 4 provides a flow diagram detailing our revised inference procedure. In response to other reviewers, our updated inference is more involved; however, revising the procedure allowed us to perform inference under several distinct parameterizations of the instability rates and compare the statistical support for each such model. As the number of parameterizations has increased, we have also added Table 1, which includes explicit definitions for each parameterization in terms of their respective parameters (also, see the new Methods subsection on “*Parameterizations of repeat instability rates*”) and statistical model comparisons. The specific parameters in each model were chosen primarily for mathematical reasons: we chose functional forms that rapidly increase with repeat tract length and are flexible enough to explore a wide range of possible instability rates (with as few parameters as possible). We would prefer not to emphasize specific interpretations for individual parameters, as they are not intended to represent explicit biological properties. Instead, interpretation of our inference results (throughout the Results and Discussion) is largely described in terms of inferred properties of the length dependent instability rates $\epsilon(L)$ and $\kappa(L)$ (e.g., the direction of expansion-contraction bias in specific length ranges). Although these qualitative properties are dependent on the range of parameter combinations that we determined to be most probable (i.e., with high posterior probabilities in our Bayesian inference), the specific meaning of each parameter is less important than qualitative properties of the length dependent rates that are most consistent with the observed distribution of repeat tract lengths.

Reviewer #3 (Remarks to the Author):

This is a very interesting manuscript arguing that repeat length distributions in hominoids and other mammals can all be explained by a unified mutational process involving expansion, contraction, and accelerated mutagenesis at long repeat length scales. I found the manuscript to be extremely careful, well explained, and informative, and I have no major complaints.

We sincerely thank the reviewer for these generous comments and for their careful reading of our manuscript.

The one thing I would suggest the authors revisit is the potential role of bioinformatic errors, which are not discussed much and can be significant for this kind of variation. In particular, homopolymer repeats are known to be especially error-prone—could these errors explain any of the difference observed between homopolymer repeats and higher order repeats?

Indeed, we agree with the reviewer that homopolymer errors in sequencing introduce a potential source of errors in our inference. Because this is related to a later question by the same reviewer, we respond to this, in detail, below.

Regarding whether this error source can generate differences in our results for mononucleotide repeats and those for longer repeated motifs, we have included only limited discussion of the latter in our manuscript. The distribution of repeat tract lengths

(referred to as the DRL in the revised manuscript) presented for each motif (see **Figure S2**) was assembled directly from the Telomere-to-Telomere genome assembly, which we believe is robust to this type of errors. We additionally compared these DRLs to those constructed from an older reference genome (e.g., the HG38 human genome assembly) and from two genomes assembled exclusively from short read sequencing (see **Figure S1B**). Importantly, we found that the shape of the DRL is virtually identical in well-populated length bins (e.g., counts >30), demonstrating the robustness of estimates of the DRL to sequencing technology (due to much larger counts than those used to estimate mutation rates). Rather than owing to sequencing errors, the observed differences in the DRLs are consistent with our understanding of differences in the mutational processes (e.g., substitution rate differences).

In contrast, estimates of per-generation instability rates generated from trio data are likely subject to homopolymer errors, along with any other motif-specific sequencing errors. These rate estimates were not used for inference in our manuscript (other than those for mono-A repeats) and should be interpreted with appropriate caution.

While not an inference, we have now included a coarse analysis in **Figure S14** that shows the rough onset lengths for repeat instability for various motifs; the onset length was quantified from the DRL for each motif (x-axis) and from the trio-based rate estimates (y-axis), which roughly agree. Although the rate estimates are susceptible to sequencing errors, this property appears to be robust to such errors, as the onset lengths appear at lengths below those subject to clear sequencing errors in our rate estimates

Additionally, based on the reviewer's suggestion, we have added several paragraphs describing systematic errors in our instability rate estimates to the Methods subsection "*Bioinformatic estimation of substitution and indel rates*".

There are large differences in estimated mutation rates between some recent papers on de novo STR mutations, e.g. Mitra, et al. 2021 doi:10.1038/s41586-020-03078-7 estimated STR mutation rates that were quite a bit higher across the board than some other studies such as Goldberg, et al. 2024 <https://doi.org/10.1093/genetics/iyae013>. Can this model speak to such differences in DNM rate estimates and which algorithms for repeat DNM calling appear to be most accurate?

Based on our estimates, as well as previous experimental evidence, repeat instability is dependent on the length of a repeat. While Mitra, et al. (2021) [6] included length-stratified rate estimates (albeit in 10 nt. bins), we believe that the reviewer is referencing the single-number estimates when comparing to rate estimates in Goldberg, et al. (2024) [7]. length dependence introduces a complication in estimating mutation rates, as averages over a range of lengths are not easily interpretable. To clarify the effect, we can slice the calculation of per-repeat instability rates into length classes. For example, estimating length-averaged expansion rate per repeat, $E[\epsilon]$ becomes an average over the DRL.

$$E[\epsilon] = \frac{\sum_{L \in [L_{min}, L_{max}]} \epsilon_L^{n_L}}{\sum_{L \in [L_{min}, L_{max}]} n_L}$$

Here, ϵ_L is the expansion rate per repeat (in any location) at length L , n_L is the number of repeats in the genome (or assembly) at length L , and the limits L_{min} and L_{max} are the minimum and maximum lengths included in the average. Note that the denominator represents the total number of repeats in this length range. The resulting estimate of the *average* expansion rate is therefore sensitive to the length dependence of both the expansion rate and the DRL weighting each length bin. Additionally, based on our own analysis, the expansion rates and counts in the DRL change quite rapidly as a function of length, making estimation of length-averaged rates particularly sensitive to the range of lengths $[L_{min}, L_{max}]$ used to compute the average expansion rate.

We attribute differences between the length-averaged rates presented in Mitra, et al. and Goldberg, et al. to be a consequence of the specific choice of filtering and averaging strategies used in each analysis, which effectively alter either the range of lengths used to average or the number of repeats counted at each length. For example, Goldberg, et al. employed rather strict read filtering requirements for quality control, which implicitly altered the accessible range of lengths included in their averaged rate estimates; this likely filtered out longer repeats preferentially, which would lower length-pooled rates (easily accounting for the roughly 4-fold reduction relative to Mitra, et al.). Further, where Mitra, et al. presented length-stratified rate estimates, these appear to be combined within each decade (e.g., 1—10, 11—20, etc.; see e.g., Extended Data Figure 2B), which effectively averages rates within each bin.

While other systematic factors may have played a role in the apparent discrepancy mentioned by the reviewer, the primary issue we see is in presenting length-averaged rates for quantities that rapidly change with repeat length. This emphasizes the importance of presenting length-stratified rate data, which we hope is clarified by our manuscript. In addition to length-stratification, we note that these authors chose to remove homopolymer repeats for specific analyses; inclusion or exclusion of individual motifs or unit lengths can alter calculations of average rates. Beyond emphasizing the importance of length-stratification, our primary insight is that a self-consistency must exist between the short- and long-timescale behavior of repeats. We thank the reviewer for pointing out that this may have bearing on future (length-stratified) instability rate estimates. We have added the following relevant text to our Discussion (final paragraph):

“[...]substantial effort continues to be dedicated to both assembling datasets and developing estimation techniques specific to repeat instability, due to the inherent difficulties associated with repetitive DNA. Given the difficulty of this task, the present work demonstrates how direct rate estimates can be informed by orthogonal data. The comparative robustness of estimates of the distribution of repeat lengths provides constraints on properties of instability that can serve as a new means for evaluating the quality of differing rate estimates.”

Conversely, how much of the parameter inferences from this paper depend on precision of the mutation rate estimates in the input trio data?

We thank the reviewer for this important question, as it prompted careful consideration of the role of systematic errors across our inference procedure. In our revised

manuscript, we have now performed a full Bayesian inference for a number of parameterizations. We have also added **Figure 4**, which provides a schematic representation of the inference procedure; in particular, there are three distinct places where we have directly incorporated empirical estimates that may be affected by sequencing errors. These include incorporation of estimated DRLs from human and other primate genomes, construction of an informative prior probability distribution from popSTR-estimated instability rate data, and inclusion of de novo estimates from pooled trio data (at and below 8 nt lengths) in the parameterized instability rates that were input into our computational model.

As stated above, estimates of the DRL appear to be more robust to errors than instability rate estimates. As part of our revised Bayesian inference procedure, we now include 36 primate DRLs constructed from their respective reference assemblies, which include a range of N50 contig lengths spanning roughly $3e4$ to $2e8$ nt (i.e., varying sequencing technologies). This ensemble of primate DRLs is used to define a rejection probability under Approximate Bayesian Computation (following the procedure in Wilkinson (2013) [1]) for each parameter combination. In doing so, we now explicitly account for model misspecification that includes the combined effects of stochastics and systematic errors (e.g., due to homopolymer sequencing errors) present in the primate DRLs. This is now described in Results “*Bayesian inference and parametric model comparison*” and in the Methods.

The second potential influence of sequencing errors lies in the construction of our informative prior. Due to qualitative differences in instability rate estimates from our pooled trio dataset and those from popSTR (see added text in Methods “*Bioinformatic estimation of substitution and indel rates*”), we chose to incorporate the latter in the form of a prior used to approximate posterior probabilities for each parameter combination. When defining the informative prior, we artificially inflated estimated statistical noise for each estimate to account for potential systematic errors. The provided posteriors that use this prior thus represent a range of potential errors in estimated rates in the intermediate length range (roughly 10–25 nt). As an alternative, we have included a naively uninformative prior (i.e., uniform probability over explored parameters), which completely excludes this set of instability rate estimates. Readers skeptical of the popSTR-based estimates may refer to these posterior probabilities for comparison (these posteriors are now provided in **Figure S9**).

Last, empirically estimated instability rates (for $L=1$ –8 nt mono-A repeats) are explicitly used in most of our parameterized mutation rate models to produce steady-state DRLs. As these rates were applied to all parameter combinations and predominantly affect counts in the lowest length bins of the DRL, any inconsistency with the empirical DRL was penalized to roughly the same degree, independent of parameter values. Further, we believe that estimated rates at very low lengths (e.g., roughly $L=1$ –5 nt) are quite reliable due to very large counts and consistency with the background indel rate. These rates remain sufficiently low such that the steady-state dynamics are dominated by substitutions alone (i.e., reach a geometric distribution at late times, independent of the specific values of instability rates for $L < 6$ nt). As instability rates increase at slightly larger lengths, they likely lose accuracy, ultimately resulting in non-monotonic behavior inconsistent with our understanding of repeat instability

(roughly at and above L=9 nt). This motivated our inclusion of these rates only up to length L=8 nt.

Estimated rates at L=8 nt play an additional role in the parameterization of primary interest, as we have defined instability rates at L=9 (the first parameterized length bin) by multiplying the L=8 expansion and contraction rates by the same parametric value m . This was done to reduce the number of parameters in our inference. The reviewer's insight was correct in pointing to the potential influence on errors in estimated mutation rates on results of our inference under this model. While our initial submission only presented this parametric model, our revised analysis includes multiple (nested) parameterizations and a formal model comparison. Comparison between parameterizations showed the strongest statistical support for the multiplier-based model. However, we have also provided complete posterior results for all models, including a four-parameter generalization of the power-law model that includes expansion and contraction rates at L=9 nt as independent parameters (see **Figure S9** for posteriors). Readers (and the reviewer) may prefer to assess the details of this four-parameter model, in which the parameterization of rates at lengths above L=8 is independent of empirically estimated rates. This four parameter model was only marginally disfavored relative to the multiplier-based model (Bayes factor ratio of 9; see **Table 1**). Importantly, the results from this larger parameterization did not change our qualitative understanding of the dynamics or interpretation of the posterior probabilities (see **Figures 5, S6** and **S9** for comparison). Accordingly, we believe that qualitative results of our inference are largely robust to misestimates of the instability rates, but specific parameter values or posterior probabilities may be quantitatively influenced by such errors. We have added the following text to Results in "*Inference of instability rates from the steady-state repeat length distribution*" to summarize these points:

"We then computed the posterior-weighted length-dependent rates of expansion and contraction for each prior and found rough consistency with popSTR-estimated rates (**Figure 5c**). One salient feature emerged, regardless of prior: expansion bias at intermediate tract lengths transitions to contraction bias at longer lengths due to the faster increase in contraction rate with length (i.e., $\tau_{\kappa} > \tau_{\epsilon}$; see **Figure 5a,c**). This likely explains the preference for the multiplier-based model, which necessarily inherits a modest initial expansion bias directly from empirical rate estimates. However, if the apparent expansion bias at L=8 is simply a consequence of homopolymer sequencing errors, correcting the direction of this bias would lead to statistical rejection of the multiplier-based model in favor of the four-parameter model (which lacks the *a priori* imposition of initial expansion bias on the parameterized rates). Regardless, inference results under the four-parameter model recapitulate the importance of a transition from expansion to contraction bias (**Figure S9**)."

As a final point, we note that we have now included one additional analysis relevant to the reviewer's question. We performed a full inference under a parameterization with no dependence on empirical instability rate estimates (i.e., the 'pure power-law' model with parameterized power-law length dependence at all lengths; see Results "*Inferring the onset of instability from the shape of the DRL*" and **Table 1**). While this toy model is undoubtedly oversimplified, specifically in poorly characterizing the rapid onset to repeat instability seen at roughly L=6-10 nt, we now show that

qualitative features like the onset length of repeat instability can be inferred directly from the shape of the DRL alone. This data-free estimate (e.g., using the uninformative prior) again suggests that our qualitative understanding of the mutational dynamics leading to the steady-state DRL is robust to systematic errors in empirical estimates of the instability rates.

Referenced publications

1. RD Wilkinson. "Approximate Bayesian computation (ABC) gives exact results under the assumption of model error", *Statistical Applications in Genetics and Molecular Biology*, vol. 12, no. 2, (2013) pp. 129-141.
2. S Kruglyak, et al. "Equilibrium distributions of microsatellite repeat length resulting from a balance between slippage events and point mutations." *Proceedings of the National Academy of Sciences* 95, no. 18 (1998): 10774-10778.
3. PW Messer, PF Arndt, and M Lässig. "Solvable sequence evolution models and genomic correlations." *Physical review letters* 94, no. 13 (2005): 138103.
4. RE Handsaker, et al. "Long somatic DNA-repeat expansion drives neurodegeneration in Huntington's disease." *Cell* 188, no. 3 (2025): 623-639.
5. MLA Hujoel, et al. "Insights into the causes and consequences of DNA repeat expansions from 700,000 biobank participants." *bioRxiv* (2024): 2024-11.
6. I Mitra, et al. "Patterns of de novo tandem repeat mutations and their role in autism." *Nature* 589, no. 7841 (2021): 246-250.
7. ME Goldberg, et al. "Effects of parental age and polymer composition on short tandem repeat de novo mutation rates." *Genetics* 226, no. 4 (2024): iyae013.